



# CH$_4$ emissions from Northern Europe wetlands: compared data assimilation approaches

Guillaume Monteil[2, 1], Jalisha Theanutti Kallingal[1], and Marko Scholze[1]

[1]Department of Physical Geography and Ecosystem Science, Lund University, Lund, Sweden
[2]Barcelona Supercomputing Center, Barcelona, Spain

**Correspondence:** Guillaume Monteil (guillaume.monteil@bsc.es)

**Abstract.** Atmospheric inverse modelling and ecosystem data assimilation are two complementary approaches to estimate CH$_4$ emissions. The inverse approach infers emission estimates from observed atmospheric CH$_4$ mixing ratio, which provide robust large scale constraints on total methane emissions, but with poor spatial and process resolution. On the other hand, in the ecosystem data assimilation approach, the fit of an ecosystem model (e.g. a Dynamic Global Vegetation Model, DGVM) to
eddy-covariance (EC) flux measurements is used to optimize model parameters, leading to more realistic emission estimates.

Coupled data assimilation frameworks capable of assimilating both atmospheric and ecosystem observations have been shown to work for estimating CO$_2$ emissions (e.g. Rayner et al. (2005)), however ecosystem data assimilation for estimation CH$_4$ emissions is relatively new. Kallingal et al. (2024) developed the GRaB-AM data assimilation system, which performs a parameter optimization of the LPJ-GUESS against eddy-covariance estimation of CH$_4$ emissions. The optimization improves
the fit to EC data, but the validity of the estimate at large scale remained to be tested.

In this study, we used the LUMIA regional atmospheric inversion system (Monteil and Scholze, 2021) to confront wetland emissions from the GRaB-AM approach to atmospheric CH$_4$ measurements in Europe. We then perform inversions using the information from GRaB-AM as prior. This let us infer a refined estimate for wetland emissions in Nordic Europe, and to explore the potential for a fully coupled data assimilation framework.

# 1 Introduction

Methane (CH$_4$) is the second most important greenhouse gas after CO$_2$, accounting for around 21% of the total effective radiative forcing of the well-mixed greenhouse gases (Forster et al., 2023). Its presence in the atmosphere has more than doubled since pre-industrialization era, with background mixing ratio at Mauna-Loa approaching the 2000 ppb (1931.91 ppbv in April 2024, according tohttps://gml.noaa.gov/ccgg/trends_ch4 (last consulted: September 2024). After a stabilization from
1998 to 2007, the atmospheric CH$_4$ concentration has started increasing again, at an accelerating pace. Although several recent studies attribute this renewed increase mainly to anthropogenic emissions (Nisbet et al., 2016; Thanwerdas et al., 2024), an important contribution from wetlands has also been proposed (Qu et al., 2022; Peng et al., 2022; Christensen, 2024). While for the global methane budget, tropical wetlands are most important, Arctic wetlands could constitute a potent positive climate



feedback (Zhang et al., 2023) and there are indications that their emissions have been increasing in recent years (Yuan et al.,
2024; Ward et al., 2024).

Emission estimates for natural wetlands can be obtained through process models, which calculate methane emissions according to various environmental inputs (meteorological forcings, soil type, hydrology, etc.). The model simulates or approximates known physical processes with various degrees of complexity. However, uncertainties on the existence or the importance of specific processes, lack of accuracy of some parameterizations, combined with the high non-linearity of the models leads to large
differences between the estimates at large scales. For example, a comparison of mean annual $CH_4$ emissions from 16 models used in the Global Carbon Project (GCP) has shown that global estimates range from 118.7 $TgCH_4$/year to 195 $TgCH_4$/year (Ito et al., 2023). Specifically for wetlands above 15°N, the estimated emissions ranged between 10.5 $TgCH_4$/year (SDGVM model) and 40 $TgCH_4$/year (ORCHIDEE). Similar ranges were also found during the WETCHIMP model intercomparison project (Melton et al., 2013), which reported a ±40% spread of the estimates around the all-model mean of 190 $TgCH_4$/year,
for global emissions.

The Global Rao-Blackwellized Adaptive Metropolis (GRaB-AM) approach developed by Kallingal et al. (2024) is a data assimilation (DA) system based on Bayesian statistics, in which parameters of the LPJ-GUESS model connected to the production, transport and oxidation of $CH_4$ pathways are adjusted to optimize the model fit to eddy-covariance flux measurements. The optimized parameters can then be used to produce a gridded estimate of the methane emissions, which combine the pro-
cess knowledge embedded in the LPJ-GUESS model with the added information from in-situ flux observations. However, the quality of the resulting emission estimate remains difficult to formally assess. The optimization is done by performing site-scale simulations, with local measurements of meteorological forcings and a good knowledge of the wetland types and their spatial distribution. Scaling up to Northern hemispheric emissions is then done using forcings from meteorological reanalysis, with hypothesis on the wetland type and fractions in each grid cell, which carry their own uncertainties. Total $CH_4$ weltands
emissions for the region north of 45°N as simulated by LPJ-GUESS are of the order of 43.09 $TgCH_4$/year for the uncalibrated (prior) model and 37.54 $TgCH_4$/year for the calibrated LPJ-GUESS model (posterior) (Kallingal et al., 2024).

An alternative approach is to infer methane emissions from their observed impact on atmospheric $CH_4$ using inverse modeling approaches (Houweling et al., 2017). Atmospheric inversions leverage the fact that atmospheric observations are sensitive to the emissions aggregated over a large area, owing to the long lifetime of atmospheric $CH_4$, and therefore can provide large-
scale constraints on methane emissions. However, this also means that the observed atmospheric methane concentrations are the result from a mixture of emissions processes. The capacity of inversions to independently constrain emissions from a specific source is therefore limited to cases where there is a distinct spatial and/or temporal emission pattern that can be used to isolate the contributions of that process to the net methane emissions. Emissions from natural wetlands dominate the methane emission budget in the arctic region, which has been used by several recent studies to derive inversion-based estimates of arctic
wetlands (Wittig et al., 2023; Tsuruta et al., 2019; Ishizawa et al., 2023).

The inverse and DA approaches assimilate complementary observations, and there could be a benefit in integrating them further in a unified $CH_4$ data assimilation system, on the model of what exists for $CO_2$ (Rayner et al., 2005). In this study, we take a step in that direction by confronting emissions estimates from the GRaB-AM data assimilation system of Kallingal et al.




(2024) to inverse modelling estimates from the LUMIA (Lund University Modular Inversion Algorithm) regional inversion

system Monteil and Scholze (2021), focusing our analysis on high-latitude European wetland emissions. The confrontation between the approaches serves both as a form of cross-validation, but also to explore the potential for a joint data assimilation setup.

## 2 Methods

We compare four main wetland emission estimates: two LPJ-GUESS, one from the default setup (LPJ-GUESS-unopt) and

one from the GRaB-AM optimized setup-GUESS (LPJ-GUESS-opt), and two LUMIA inversions, using these LPJ-GUESS simulations, and their uncertainties as prior: LUMIA-Lprior (using LPJ-GUESS-unopt as prior) and LUMIA-Lpost (using LPJ-GUESS-opt as prior). These four simulations correspond respectively to a pure bottom-up estimate, a flux-observation informed estimate, a atmospheric observation informed estimate, and an estimate informed both by flux and atmospheric data. Two additional LUMIA inversions were performed as sensitivity simulations (LUMIA-Lprior+corr, LUMIA-Lpost+corr), with

a different prior error covariance structure (see also Section 3.1).

The study covers the year 2018, for the LUMIA inversion domain represented in Figure 1, although we focus mainly on the observations in the Nordic sub-domain (red box in Figure 1). The domain extent is based on that of a $CH_4$ regional inverse modelling intercomparison, jointly organised by WMOs Integrated Global Greenhouse Gas Information System (IG3IS) and the Horizon-Europe CoCO2 project (https://coco2-project.eu, last consulted: 23/09/2024), to which this study contributes.

### 75 2.1 Wetland emissions modelling

#### 2.1.1 LPJ-GUESS

LPJ-GUESS is a dynamic global vegetation model (DGVM), designed to simulate the interactions between vegetation, soil, and their responses to environmental changes and management (Sitch et al., 2003; Smith et al., 2001, 2014). The model can simulate vegetation dynamics, carbon and water cycles, and soil biogeochemistry from local to global scales, including the

simulation of methane fluxes from natural wetlands.

For this study, we used the Arctic-enabled version 4.1 of the model (Smith et al., 2014), which differs from previous versions for having detailed representation of wetland $CH_4$ emission. The process descriptions of the $CH_4$ module were mostly adopted from the LPJ-WHyMe model (Wania et al., 2010), and are described in detail in (McGuire et al., 2012). It is based on a "potential carbon pool", which is then decomposed to soil organic carbon distributed vertically in the soil layers. Methanogens

use this decomposed organic carbon and produce $CH_4$. A part of this produced $CH_4$ gets oxidized by $O_2$ and the remainder is transported to the atmosphere by diffusion, ebullition, or plant-mediated transport (see Wania et al. (2010); Kallingal et al. (2024) for more details). The model is driven by daily climate data including air temperature, precipitation, and shortwave radiation taken from the Climatic Research Unit-Japanese Reanalysis (CRU-JRA; Harris et al. (2020) dataset. Annual atmospheric $CO_2$ concentrations, as additional model input for LPJ-GUESS, are obtained from the Global Monitoring Laboratory



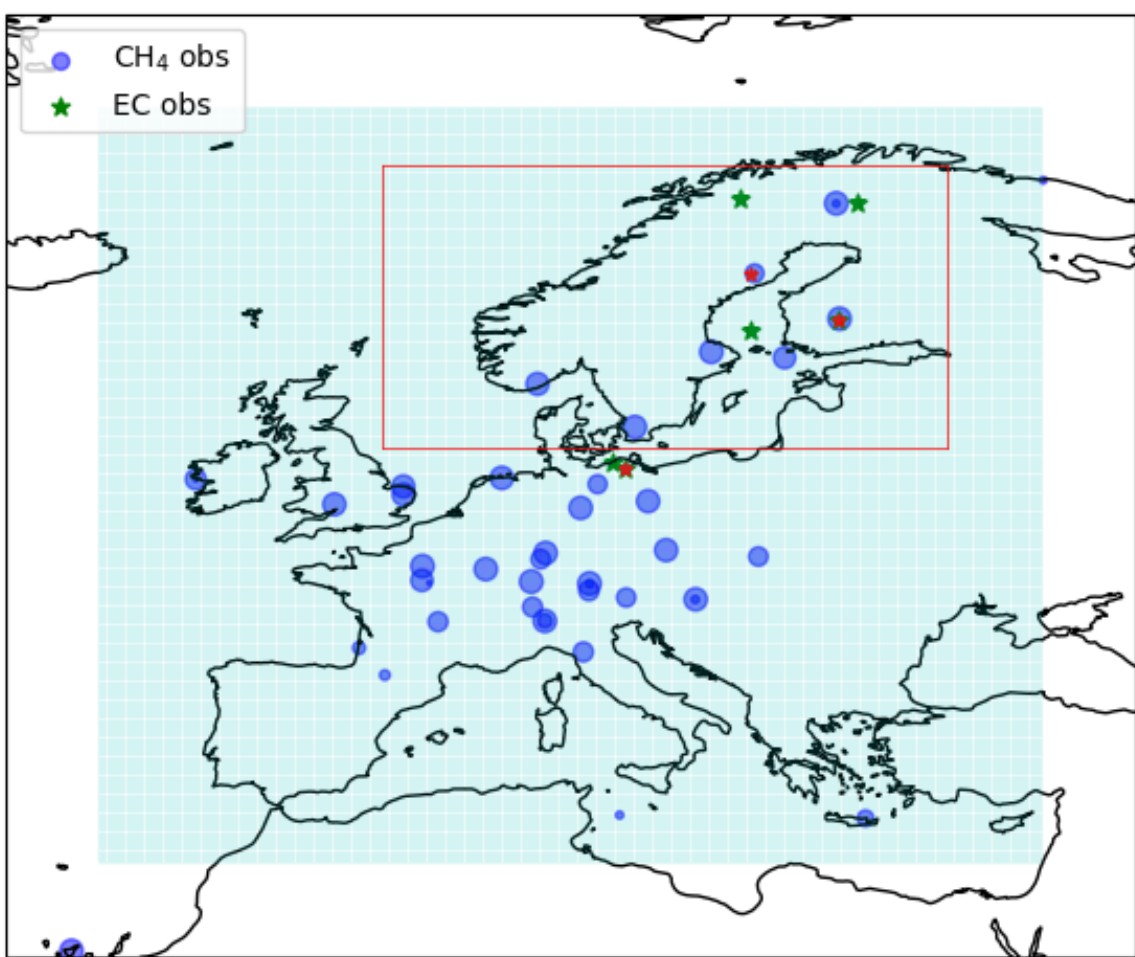

**Figure 1.** LUMIA inversion domain (cyan grid), the position of the observations used in the LUMIA inversions (blue dots, with the area proportional to the number of assimilated observations); position of the eddy-covariance sites used in the GRaB-AM optimization (green stars) and Nordic domain of interest (red box). The red dots mark the position of the sites used in Section 3.3.1.



(https://gml.noaa.gov/ccgg/trends), and the soil property data was extracted from WISE5min, V1.2 Soil Property Database (Batjes, 2005).

Model simulations covering the area north of 45°N are produced using PEATMAP (Xu et al., 2018), which combines geospatial information from various sources to create a global map of wetland extent. PEATMAP has been used in several studies mainly because it is an updated quantification of peat land extend, and it focuses on mapping peatlands, such as

marshes and swamps, which are the dominant wetlands in northern latitudes (Peltola et al., 2019; Aalto et al., 2024; Müller and Joos, 2020).

### 2.1.2 GRaB-AM flux data assimilation framework

To optimize the methane module of LPJ-GUESS, Kallingal et al. (2024) developed the GRaB-AM data assimilation framework, which seeks to optimize the value of ten highly sensitive parameters in the methane module of LPJ-GUESS (connected

to production, transport and consumption pathways of $CH_4$), based on the model fit to eddy-covariance (EC) flux observations. The minimization is performed using an adaptive scheme of the Markov Chain Monte Carlo Metropolis-Hastings algorithm (Metropolis et al., 1953; Hastings, 1970). In Kallingal et al. (2024), observations from 14 natural wetlands distributed across the Northern Hemisphere above 40° and over a total period of 20 years (from 2000 to 2020 with individual sites contributing observations over different years within this time period) were assimilated. The number of sites used for the GRaB-AM

optimization exceeds the indicated sites shown in Figure 1 (a full list of the sites is given in Kallingal et al. (2024)).

In this study we computed two ensembles of hundred gridded LPJ-GUESS simulations each, randomizing the values of the LPJ-GUESS parameters adjusted by the GRaB-AM algorithm. In the first ensemble (LPJ-GUESS-unopt) the parameter values were drawn based on their prior estimate and standard deviations of 40% of their assumed ranges. In the second ensemble (LPJ-GUESS-opt), the ensemble members were drawn from the 90% confidence interval (1.645 time the standard deviations)

of the posterior distribution after the burn-in period of the MCMC chain.

### 2.2 Atmospheric inverse modelling

The consistency of $CH_4$ emission from LPJ-GUESS (LPJ-GUESS-unopt and LPJ-GUESS-opt) with atmospheric $CH_4$ mixing ratio measurements was tested using the LUMIA inversion framework. The comparison requires using an atmospheric transport model, and accounting for contributions of other methane sources, and from lateral boundary conditions. The model data

mismatches then serve to infer a further correction to the emission estimates.

### 2.2.1 Inversion approach

LUMIA Monteil and Scholze (2021) is a regional atmospheric inversion setup developed initially to estimate European $CO_2$ inversions using in-situ concentration measurements such as those provided by the ICOS network (Monteil et al., 2020; Munassar et al., 2023; Gómez-Ortiz et al., 2023). This study is the first application to a non-$CO_2$ tracer.



The inversion seeks to determine the set of regional $CH_4$ emissions that is the most consistent with a dataset of observed in-situ $CH_4$ mixing ratios. The impact of emissions outside the regional domain is provided through a prescribed background term. The link between emissions and volume mixing ratio given by:

$$\mathbf{y} + \varepsilon_\mathbf{o} = \mathbf{Hx} + \mathbf{y}_{bg} + \varepsilon_\mathbf{m} \tag{1}$$

with $\mathbf{y}$ the observations vector contains observations of the atmospheric $CH_4$ mixing ratio, and $\mathbf{x}$ the control vector con-
tains the variables that we seek to optimise: in our case, the daily $CH_4$ emissions, grouped in two categories (wetlands and non-wetlands), at a 0.25° resolution over a regional domain ranging from -15°W, 33°S to 35°W, 73°N (Figure 1). The regional transport operator $\mathbf{H}$ contains the sensitivity of the observations $\mathbf{y}$ to the (regional) emissions $\mathbf{x}$ ($\mathbf{H}_{i,j} = \partial \mathbf{y}_i / \partial \mathbf{x}_j$). The background concentrations ($\mathbf{y}_{bg}$) are provided as timeseries of baseline concentrations directly at the observation sites, following the two step approach of Rödenbeck et al. (2009) (see Section 2.2.3). The error terms $\varepsilon_\mathbf{o}$ and $\varepsilon_\mathbf{m}$ represent respectively the
measurement error and the model error.

The optimal control vector $\hat{\mathbf{x}}$ is given as the one that minimises a cost function $\mathcal{J}(\mathbf{x})$, such that

$$\mathcal{J}(\mathbf{x}) = \frac{1}{2} (\mathbf{x} - \mathbf{x_b})^T \mathbf{B}^{-1} (\mathbf{x} - \mathbf{x_b}) + \frac{1}{2} (\mathbf{Hx} + \mathbf{y}_{bg} - \mathbf{y})^T \mathbf{R}^{-1} (\mathbf{Hx} + \mathbf{y}_{bg} - \mathbf{y}) \tag{2}$$

The first part evaluates the goodness of fit of the estimated control vector $\mathbf{x}$ to its prior estimate $\mathbf{x_b}$, normalised by the prior error-covariance matrix $\mathbf{B}$, which contains the uncertainty of $\mathbf{x_b}$. The right hand side term evaluates the model fit to the
observations $\mathbf{y}$, normalised by the observation error-covariance matrix $\mathbf{R}$, which combines the model ($\varepsilon_\mathbf{m}$) and measurement ($\varepsilon_\mathbf{o}$) uncertainties (i.e. it is the total uncertainty on the model-data mismatch).

The optimal control vector $\hat{\mathbf{x}}$, which satisfies $\nabla_x \mathcal{J}(\hat{\mathbf{x}}) = 0$, is the set of $CH_4$ emissions that represents the best compromise between fitting the observations and limiting the departures from the prior estimate (which implicitly carries the knowledge of the models and data used to construct the prior emissions).

The solution is searched for iteratively, using a conjugate gradient algorithm provided by the python "scipy.optimize" package (which employs a nonlinear conjugate gradient algorithm by Polak and Ribiere, a variant of the Fletcher-Reeves method described in (https://docs.scipy.org/doc/scipy/reference/generated/scipy.optimize.minimize.html#rdd2e1855725e-5) pp.120-122). This solver is not optimal for our setup (a linear CG algorithm would be better suited, since our optimization problem is strictly linear), but turned out to be more practical and I/O efficient than the Lanczos (1952) linear solver used in previous LUMIA
papers (e.g. Monteil et al. (2020); Munassar et al. (2023)), while giving qualitatively equivalent results.

### 2.2.2  Regional transport model

The regional transport operator $\mathbf{H}$ was computed using the FLEXPART 10.4 Lagrangian particle dispersion model. FLEXPART is not called directly within the inversion, but is used before, to pre-compute observation footprints, i.e. rows of the $\mathbf{H}$ matrix from Equation 2. These footprints are stored on disk and simply read during the successive phases of the inversion. Each





footprint was obtained by simulating the dispersion, backward in time starting from the observation time and position, of ten thousand virtual air particles, based on meteorological fields from the ECMWF ERA5 reanalysis (at a 0.25°, hourly resolution), and limited to the aforementioned European domain: the particles are destroyed when they reach the edge of the domain. The aggregated residence time of the particles near the surface (below 100 m above ground) is used as a proxy for the sensitivity of the observations to the emissions.

### 2.2.3 Boundary conditions

The background vector $\mathbf{y}_{bg}$ accounts for all the contributions to the observed $CH_4$ mixing ratio that are not adjusted by the inversions: the impact of the initial condition, the impact of $CH_4$ emissions from outside the regional domain, the impact of European $CH_4$ emissions having left the domain on the $CH_4$ mixing ratio of air masses (re-)entering it, and the impact of the various $CH_4$ sinks (reactions with OH, and, in the stratosphere, with Cl and $O(^1D)$).

The background concentrations were taken from the CAMS v19r1 global $CH_4$ reanalysis (Segers, 2020), which relies on the TM5-4DVAR global atmospheric inversion system. The concentrations baselines were extracted using the two-step scheme of Rödenbeck et al. (2009) (see also Bergamaschi et al. (2022) for the CAMS implementation, and Monteil and Scholze (2021) for the usage in LUMIA). These baselines were provided as part of the CoCO2 $CH_4$ inversion intercomparison, therefore they are purely an external input to our modeling setup.

### 2.2.4 Prior emissions and uncertainties

The inversions solve for $CH_4$ emissions grouped in two "super-categories": wetland and non-wetlands. The latter groups the contributions of all remaining categories, both anthropogenic (mainly agriculture, waste management and fossil fuel emissions), and natural (geological emissions, termites, lakes and oceans). Anthropogenic emissions are taken from the EDGAR v6.0 emission inventory (Crippa et al., 2019), wetland emissions are taken from the LPJ-GUESS simulations described in Section 170 2.1.1, with or without the parameters optimization described in Section 2.1.2 (depending on the simulation). Natural emissions are taken from various climatological estimates, reported in Table 1. All the emissions were regridded from their original resolution to the 0.25°, daily resolution of the FLEXPART footprints.

The spatial distribution of the emissions is shown in Figure 3, while their temporal distribution is shown in Figure 2. Wetland emissions are the only category that has exhibits a strong seasonality (biomass burning emissions are seasonal as well, but very 175 low overall, so their influence on the observations is negligible). There is also a geographical separation between emissions from wetlands, which are concentrated in Northern Europe, and emissions from the other categories, which are more significant in the rest of the continent, and tend to overlap in time and space. This, and the fact that the observation network is relatively dense in Northern Europe (Figure 1), where wetland emissions are important, justify resolving wetland emissions separately from the other $CH_4$ sources in the inversions.

The emission uncertainties are stored in the prior error-covariance matrix $\mathbf{B}$, which is composed, for each category $c$, of four components: the vector $\sigma_{\mathbf{x_c}}$ containing the standard deviations of the emission themselves, two correlation matrices, $C_c^h$ and $C_c^t$, storing respectively the prescribed correlations in the spatial and in the temporal dimension, and a scalar scaling factor





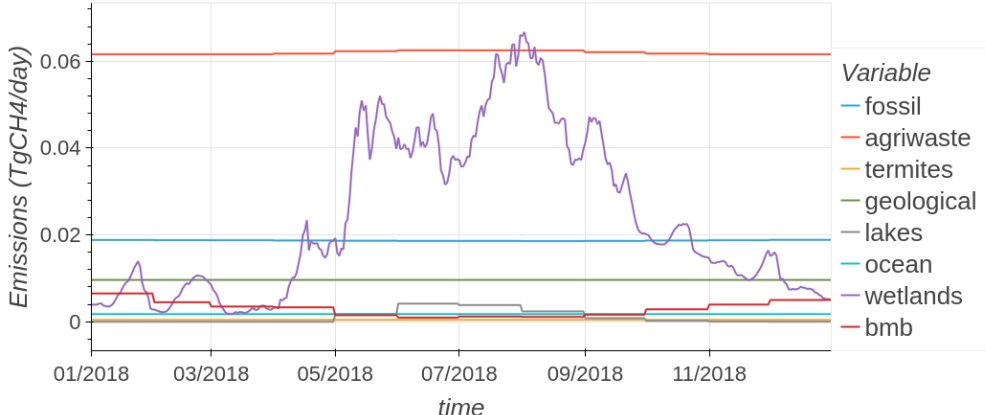

**Figure 2.** Prior CH$_4$ emissions used in the "LUMIA-Lprior" inversion

**Table 1.** Methane (prior) emissions used in the LUMIA inversions.

| Category | Annual total (TgCH$_4$) | Source | Temporal resolution | Spatial resolution | Climatological |
|---|---|---|---|---|---|
| Wetlands | 5.5-8.7 | LPJ-GUESS (This study) | daily | 0.5° | |
| Agriculture and waste | 22.6 | EDGAR v6.0 (Crippa et al., 2019) | monthly | 0.1° | |
| Fossil | 6.8 | EDGAR v6.0 (Crippa et al., 2019) | monthly | 0.1° | |
| Biomass burning | 1.1 | GFED-4.1s (Randerson et al., 2017) | 0.25° | monthly | |
| Oceans | 0.6 | Weber et al. (2019) | 0.25° | monthly | yes |
| Inland water | 0.4 | Johnson et al. (2022) | 0.1° | monthly | yes |
| Geology | 3.5 | Etiope et al. (2019) | - | annual | yes |
| Termites | 0.2 | Saunois et al. (2020) | - | annual | yes |

$\gamma_c$, which is used to enforce a specified total annual uncertainty. No cross-category correlations are assumed, therefore the error-covariance matrix is written, for each category, as:

$$\mathbf{B_c} = (C_c^t \otimes C_c^h)^T \sigma_{\mathbf{x_c}}{}^T \gamma_c^2 \sigma_{\mathbf{x_c}} (C_c^t \otimes C_c^h) \tag{3}$$

For wetland emissions, the standard deviations $\sigma_{\mathbf{w}}$ are directly given by the standard deviation of the LPJ-GUESS ensembles (Sections 2.1.1), whereas for non-wetland emissions, they are set proportional to the absolute value of the emissions. Note that only their relative values matter, since the total uncertainty is determined by $\gamma_c$. The inversion-specific values used for $\gamma$, $C^t$ and $C^h$ are provided in Section 3.1.

**2.2.5 Observation and observational uncertainties**

The LUMIA inversions were constrained by in-situ observations from 43 European in-situ and flask measurement sites, from various observation networks (see Table 2 and Figure 1), most of which are now part of the ICOS network of in-situ measure-



**Figure 3.** Prior CH$_4$ emission maps from natural wetlands, fossil fuels, agriculture and waste and "natural" sectors. The latter groups together emissions from lakes and oceans, geological sources, biomass burning and termites, but is largely dominated by the geological emissions.

ments. The observation frequency is typically hourly, but we filtered the observations to avoid assimilating observations close to the transition between the planetary boundary layer (PBL) and the free , as this is where model errors on the PBL height would have the largest impact. For most sites, afternoon data was selected (from 11:00 to 17:00, local time), when the PBL is expected to be the most developed. For high altitude sites (above 1000 m altitude a.m.s.l), night time data was used instead (from 0:00 to 4:00, local time), when the observations are expected to be well above the PBL. At a few sites (Hohenpeissenberg, Hegyhatsal, Ispra, Mace Hear and Pallas), there are also a some observations from flask measurements, for these, no specific filter was applied.

The uncertainties (diagonal of **R** in Equation 2) are set as the quadratic sum of the measurement uncertainty $\varepsilon_{obs}$ and of the model uncertainty $\varepsilon_{mod}$. The measurement uncertainty is taken from the observation datasets (when available), with a





minimum and default value of 20 ppb. The model uncertainty should in theory be set close to the random component of the
error that the model would make when simulating the observations based on the "true" emissions, and excluding systematic
component of that error. In practice, that true error is unknown. Here we constructed it as site-specific weekly uncertainty,

based on the mismatch between the observed and modelled short term $CH_4$ variability. The procedure is conducted for each
site, in four steps:

1. Compute the prior model estimate for the observations, $\mathbf{y}_{apri}$, corresponding to the prior emissions described in Section
   2.2.4.

2. Separate the modelled ($\mathbf{y}_{apri}$) and observed ($\mathbf{y}$) time series into baselines and anomalies. The baselines are computed as
weekly rolling weighted averages (with the inverse of the measurement uncertainties used as weights), and the anomalies
   are obtained by subtracting these baselines from their respective timeseries.

3. Compute the standard deviation ($\sigma_{mod}^{site}$) of the difference between the anomalies in modelled (prior) and observed $CH_4$
   mixing ratios.

4. The model uncertainty of a single observation is finally given by $\varepsilon_{mod}^{i} = \sigma_{mod}^{site} \times \sqrt{n_{obs}}$, where $n_{obs}$ is the number of
observations in a $\pm 3.5$ days interval surrounding the observation $i$.

The rationale behind this approach is that the inversions should be able to efficiently reduce the mismatch between the
baselines by adjusting the $CH_4$ emissions, but will likely struggle more with reducing the model-data mismatches between the
modelled and observed sub-weekly variability. In practice this results in a larger uncertainty at sites close to large $CH_4$ emitters,
such as Ispra, Saclay and Norunda, reducing their relative weight in the inversion. The last step (4) ensures that the weight of
a site doesn't depend on the observation frequency (if there are more observations within a given week, the individual weight
of each observation will be reduced accordingly).

## 3 Results

The LUMIA inversions use wetland emission estimates and uncertainties computed in the two LPJ-GUESS ensembles. We
therefore first present the results from these two ensembles, then compare the four $CH_4$ emission estimates and analyze their
consistency with observations.

### 3.1 Model-derived $CH_4$ emission uncertainties

The prior error-covariance matrix in LUMIA ($\mathbf{B}$ in Equation 2) is constructed, for each emission category, based on three
components (see Section 2.2.1):

1. an estimation of the error correlations (in the form of a pair of temporal and spatial correlation matrices),

2. a vector of (normalized) prior uncertainties,





**Table 2.** Observation time series assimilated in the LUMIA inversions. The number of observations assimilated is reported in the "nobs" column. The "Model error" column shows the assumed model representation error in the LUMIA-Lprior inversion (the numbers differ slightly in the LUMIA-Lpost inversion, since they are calculated based on the prior fit to the data). When available, the DOI or PID of the data are shown in their corresponding entry in the bibliography.

| code | Station | Latitude | Longitude | Elevation | Inlet height | Reference | nobs | Model error (ppb) |
|---|---|---|---|---|---|---|---|---|
| bir | Birkenes, Norway | 58.39 | 8.25 | 215 | 3 | Lunder and Platt | 3562 | 13.77 |
| bis | Biscarosse, France | 44.38 | -1.23 | 73 | 47 | Lopez and Ramonet (2024) | 339 | 3.96 |
| cmn | Mt Cimone, Italy | 44.17 | 10.68 | 2165 | 7 | Arduini | 2025 | 11.81 |
| fkl | Finokalia, Greece | 35.34 | 25.67 | 150 | 15 | Delmotte et al. (2024b) | 970 | 12.38 |
| hei | Heidelberg, Germany | 49.42 | 8.68 | 113 | 30 | Hammer and Levin (2024) | 3784 | 23.80 |
| hpb | Hohenpeissenberg, Germany | 47.80 | 11.01 | 934 | 131 | Kubistin et al. (2024b) | 3899 | 25.53 |
| hpb | Hohenpeissenberg, Germany | 47.80 | 11.02 | 936 | 5 | Lan et al. | 49 | 3.59 |
| htm | Hyltemossa, Sweden | 56.10 | 13.42 | 115 | 150 | Heliasz and Biermann (2024) | 3857 | 16.59 |
| hun | Hegyhatsal, Hungary | 46.95 | 16.63 | 248 | 96 | Lan et al. | 47 | 26.14 |
| hun | Hegyhatsal, Hungary | 46.96 | 16.65 | 248 | 96 | Haszpra | 3772 | 1.68 |
| ipr | Ispra, Italy | 45.81 | 8.64 | 210 | 16 | Scheeren et al. (2007) | 3845 | 32.32 |
| jfj | Jungfraujoch, Switzerland | 46.55 | 7.99 | 3570 | 10 | Steinbacher | 1999 | 7.77 |
| kas | Kasprowy Wierch, Poland | 49.23 | 19.98 | 1987 | 2 | Chmura et al. (2024) | 1783 | 14.88 |
| kre | Kresin u Pacova, Czech Republic | 49.57 | 15.08 | 534 | 250 | Marek et al. (2024) | 3696 | 13.63 |
| lin | Lindenberg, Germany | 52.17 | 14.12 | 73 | 98 | Kubistin et al. (2024c) | 3747 | 15.50 |
| lmp | Lampedusa, Italy | 35.51 | 12.61 | 45 | 5 | Lan et al. | 47 | 14.51 |
| lut | Lutjewad, Netherlands | 53.40 | 6.35 | 1 | 60 | Chen and Scheeren (2024) | 3859 | 43.66 |
| mhd | Mace Head, Ireland | 53.31 | -9.90 | 5 | 21 | Lan et al. | 5 | 1.19 |
| mhd | Mace Head, Ireland | 53.33 | -9.90 | 5 | 0 | Prinn et al. (2018) | 2467 | 6.62 |
| nor | Norunda, Sweden | 60.09 | 17.48 | 46 | 100 | Lehner and Mölder (2024) | 3859 | 31.52 |
| ope | Observatoire pérenne de l'environnement, France | 48.56 | 5.50 | 390 | 120 | Ramonet et al. (2024a) | 3835 | 17.10 |
| pal | Pallas, Finland | 67.96 | 24.11 | 565 | 5 | Lan et al. | 32 | 12.97 |
| pal | Pallas, Finland | 67.97 | 24.12 | 560 | 7 | Hatakka | 3925 | 50.61 |
| pdm | Pic du Midi, France | 42.94 | 0.14 | 2877 | 10 | Delmotte et al. (2024a) | 148 | 4.33 |
| puy | Puy de Dome, France | 45.77 | 2.97 | 1465 | 10 | Colomb et al. (2024) | 2009 | 12.73 |
| rgl | Ridge Hill, United Kingdom | 52.00 | -2.54 | 204 | 90 | O'Doherty et al. (2024) | 3668 | 16.59 |
| sac | Saclay, France | 48.72 | 2.14 | 160 | 100 | Ramonet et al. (2024b) | 3903 | 38.51 |
| smr | Hyytiala, Finland | 61.85 | 24.29 | 181 | 125 | Levula and Mammarella (2024) | 3883 | 25.53 |
| ssl | Schauinsland, Germany | 47.90 | 7.92 | 1205 | 6 | Meinhardt | 3959 | 14.42 |
| tac | Tacolneston Tall Tower, United Kingdom | 52.52 | 1.14 | 56 | 185 | O'Doherty and Pitt (2024) | 3620 | 17.80 |
| toh | Torfhaus, Germany | 51.81 | 10.54 | 801 | 147 | Kubistin et al. (2024a) | 3841 | 20.98 |
| trn | Trainou, France | 47.96 | 2.11 | 131 | 180 | Ramonet et al. (2024c) | 3178 | 17.72 |
| uto | Uto, Baltic Sea | 59.78 | 21.37 | 8 | 57 | Hatakka and Laurila (2024) | 3151 | 12.40 |
| wao | Weybourne, United Kingdom | 52.95 | 1.12 | 10 | 0 | Forster and Manning (2024) | 3610 | 22.19 |
| zsf | Zugspitze-Schneefernerhaus, Germany | 47.42 | 10.98 | 2667 | 3 | Couret and Schmidt (2024) | 2142 | 9.08 |





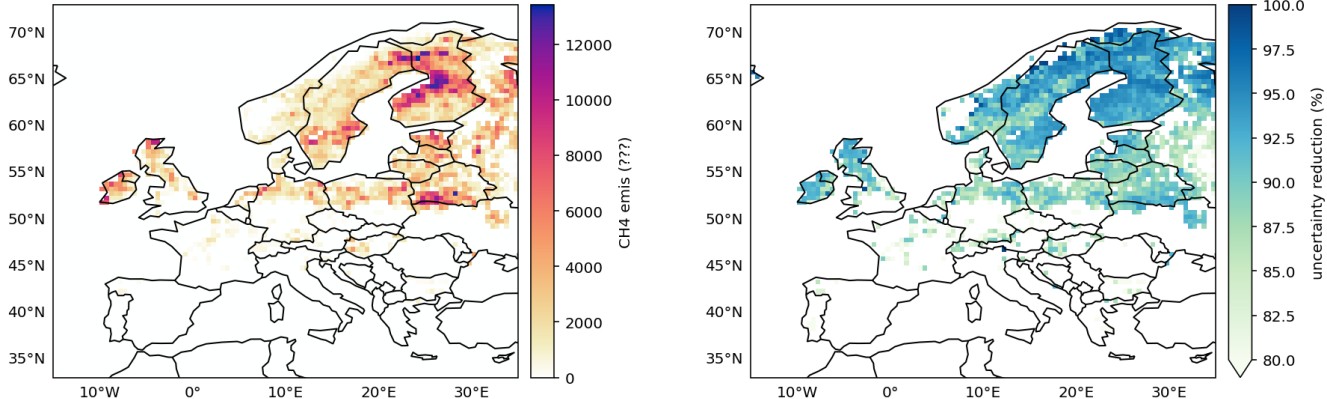

**Figure 4.** Wetland emission uncertainties, as per the LPJ-GUESS-unopt ensemble (left) and percentage uncertainty reduction in the LPJ-GUESS-opt ensemble (right).

3. a total, category-specific annual uncertainty estimate

The emission uncertainties corresponding to the LPJ-GUESS-unopt and LPJ-GUESS-opt simulations were estimated by through two ensemble simulations of 100 members each (see Section 2.1.2). The ensemble standard deviation drops from 6.40 TgCH$_4$/year in LPJ-GUESS-unopt to 0.45 TgCH$_4$/year in LPJ-GUESS-opt. In the LPJ-GUESS-unopt ensemble, the un-
certainties are concentrated in regions with strong CH$_4$ emissions: Northern Finland, Scandinavian Arctic, Southern Sweden, Southern Poland and North-West coasts of Ireland and Scotland (Figure 4). The GRaB-AM optimization reduces the uncertainties everywhere, but predominantly at high-latitudes, with the strongest reduction ($\approx$-95%) being obtained in the Nordic region.

The error correlations are arguably more important for the LUMIA inversions: large error correlations effectively reduce
the dimensionality of the problem, making it in turn easier to resolve the contributions from separate categories. Full error covariance matrices can be computed from the ensemble but are too large to fit in memory and be of practical use in the inversions. Instead, in Figure 5, we show the average error correlations as function of distance (in space and time).

The error correlations are generally larger in the prior ensemble (LPJ-GUESS-unopt) than in the posterior one. In the spatial dimension, there is a lot of variability, but overall, there is a very rapid drop in correlation values, which stabilize around 0.55
in LPJ-GUESS-unopt and around 0.35 in LPJ-GUESS-opt, after approximatively 250 km. The correlations decline further with increased distance, but at a very slow pace. Temporally, correlations decrease almost instantly in LPJ-GUESS-unopt, but remain in a 0.55-0.65 range after that, whereas they decrease more gradually in LPJ-GUESS-opt, reaching below 0.4 after 200 days.

The interpretation of spatial correlations is further complexified by the fact that the number of active CH$_4$ emission grid cells
is not constant throughout the year, and therefore the correlation-distance relationship is not constant. For computing Figure 5, we ignored the time dimension for the spatial correlations plot, and the space dimension for the temporal correlation plot





(therefore, the averaged correlation for two points distant by e.g. 500 km includes the correlation between emission components at different times of the year). This somewhat mimics the way the prior error-covariance matrix $\mathbf{B}$ is constructed in LUMIA (i.e. with correlations based on as a Kronecker product of spatial and temporal correlation matrices).

We performed an ensemble of sensitivity tests to determine the most appropriate formulation for the error-covariances in LUMIA. The two main inversions, LUMIA-Lprior and LUMIA-Lpost, use the wetland error distributions ($\sigma_{\mathbf{x}}$ in Equation 3), but correlation matrices based on more traditional exponential correlation-decay functions ($corr(x) = e^{-x/L}$), with correlation lengths $L_h$ of 1000 km in $C_h$ (shown in Figure 5), and $L_t = 30$ days in $C_t$, and their annual uncertainty $\gamma_{wetland}$ was set to 0.5 TgCH$_4$/year.

In addition, two sensitivity inversions were computed, LUMIA-Lprior+corr and LUMIA-Lpost+corr, which take their annual uncertainty ($\gamma_{wetland}$) directly from the standard deviation of the annual emissions in their corresponding LPJ-GUESS ensemble, and use the ensemble-based correlation-distance relationships shown in Figure 5. The results were very similar in the Nordic region of interest, therefore for most of the analysis, we choose to rely on LUMIA-Lprior and LUMIA-Lpost. This also acknowledges the fact that ensemble-derived uncertainty estimates ignore errors from the driving data of LPJ-GUESS, and from the processes incorrectly modelled in it. Also, the spatial correlations in the ensembles are not constant through time, therefore the decomposition in a spatial and a temporal correlation matrices is not a very accurate approximation of the actual correlations of the ensemble. Finally, setting the annual uncertainty to the same value in both inversion facilitates the interpretation of the results.

## 3.2   CH$_4$ emission

### 3.2.1   Annual CH$_4$ emissions

The non-optimized LPJ-GUESS model (LPJ-GUESS-unopt) points to an emission total of 8.7 TgCH$_4$/year, including 4.3 TgCH$_4$/year in the Nordic sub-domain. The GRaB-AM optimization suggests lowering these values to 5.5 TgCH$_4$ (-37%) and 2.5 TgCH$_4$ (-42%), respectively. The two LUMIA inversions point to further reductions, down to 4.2 TgCH4/year, including 1.3 TgCH4/year in the Nordics, in LUMIA-Lprior (respectively -52% and -70% compared to LPJ-GUESS-unopt), and down to 3.0 TgCH4/year (-65%) with 1.1 TgCH4/year (-74%) in the Nordics, in LUMIA-Lpost (Figure 5). The two sensitivity inversions also lead to very similar results (Figure 6).

In contrast, the inversions lead to much lower adjustments to non-wetland emissions, both in relative and absolute terms. The priors (i.e. LPJ-GUESS-unopt and LPJ-GUESS-opt in Figure 6) are 35 TgCH$_4$ for the full domain, and 35.3 TgCH4/year (+0.85%) and 36 TgCH4/year (+2.9%) respectively in LUMIA-Lprior and LUMIA-Lpost. The contribution of the Nordic region to this is very small, with 3.3 TgCH$_4$/year in the prior (comparable in magnitude to the wetland emissions in that region). The inversions reduce these to 2.4 TgCH$_4$/year (-27%) and 2.6 TgCH$_4$/year (-21%), respectively in LUMIA-Lprior and LUMIA-Lpost. Here again, the difference between the reference inversions and their sensitivity run counterparts is insignificant.





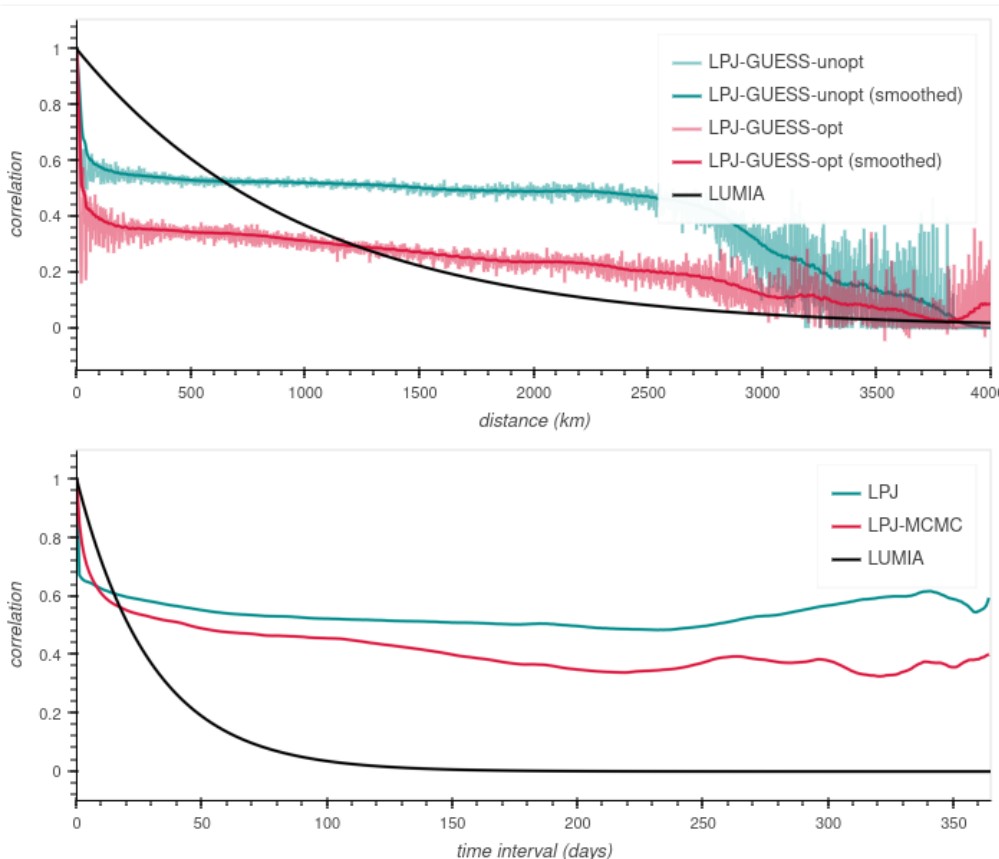

**Figure 5.** Spatial (top) and temporal (bottom) correlation-distance relationships for the two LPJ-GUESS ensembles. The black line represents the correlation settings used in the LUMIA simulations.

### 3.2.2 Seasonal cycle of the wetland emissions

Temporal emission adjustments are shown in Figure 7 (note the different y-ranges in the figure). For the figure clarity, results from the LUMIA-Lprior+corr and LUMIA-Lpost+corr inversions are shown in SI Figure 1.

For wetlands, the patterns are very similar between the full-domain and the Nordic region (which reflects the fact that this region accounts for more than half of the European wetland emissions). In LPJ-GUESS-unopt, the emissions remain close to zero in the first quarter of the year, except for two small peaks at the end of January and of March. The emissions then follow a

"double-peak" pattern, with a first peak around late may, particularly pronounced in the Nordics, and the main peak in August, after which the emissions decline steadily to reach nearly zero at the end of the year.

The assimilation of in-situ flux data in LPJ-GUESS-opt leads to roughly a halving of the emissions, mainly during the May to October period. Within the Nordic region, the May emission peak is almost fully preserved, whereas the remaining





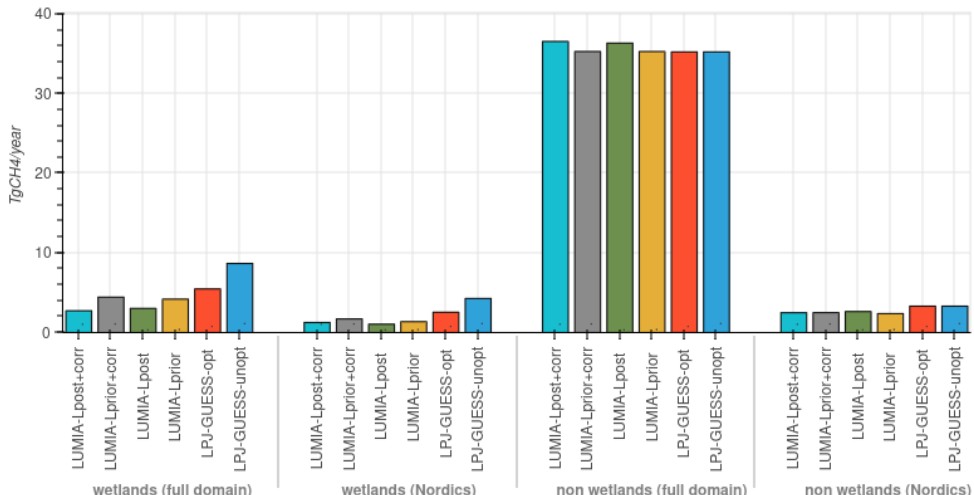

**Figure 6.** Annual emission estimates (in TgCH$_4$/year) for the wetland and non-wetland emission categories. LPJ-GUESS-unopt and LPJ-GUESS-opt are respectively the priors of LUMIA-Lprior(+corr) and LUMIA-Lpost(+corr).

part of the summer variability is smoothed. Outside the Nordics, the temporal structure of the emissions is better preserved.
LPJ-GUESS-opt points to a reduction of the amplitude of the emission peaks in January-February (mostly visible outside the Nordics). On the other hand, the emissions after October remain similar to LPJ-GUESS-unopt.

The two main LUMIA inversions lead to very consistent results within the Nordic region, only with more short-term variability in LUMIA-Lprior. Both inversions point to a nearly complete reduction of the May emission peak found in the LPJ-GUESS simulations, and to summer emissions further reduced compared to LUMIA-Lpost, especially after September.

Outside the Nordics, LUMIA-Lpost leads to a further reduction of the emissions compared to LPJ-GUESS-opt (i.e. its prior), except during June-July where the two simulations are nearly identical. On the other hand, the emissions in LUMIA-Lprior closely follow those in LPJ-GUESS-opt from March to August, which is remarkable because these two emission products have been optimized using two independent techniques, using two completely independent observation datasets.

The two sensitivity inversions LUMIA-Lprior+corr and LUMIA-Lpost+corr lead to comparable results, on multi-day aver-
ages. However, LUMIA-Lprior+corr displays an extremely high short-term variability (much more that LPJ-GUESS-unopt, its prior). We hypothesize that this is due to the very high uncertainty on wetland emissions used in that simulation (6.4 TgCH$_4$, i.e. $\approx$ 13 times more than in the other inversions), which, associated to the fact that LPJ-GUESS-unopt tends to alternate (at the grid cell level) between days with strong emissions and days with near zero emissions, makes that inversion very under-constrained. However, on a multi-day average, it is remarkably similar to LPJ-GUESS-opt within the Nordic region, until
August, and to LUMIA-Lprior for the rest of the year, and also in the rest of Europe. The difference between LUMIA-Lpost and LUMIA-Lpost+corr is mainly in extra-Nordic wetlands, where LUMIA-Lpost+corr infers a further reduction of the estimate. This is a consequence of the longer spatial correlations in that inversion (see Figure 5), which propagate the emission adjustments over larger distances.





### 3.2.3 Temporal variability of the other emissions

The temporal adjustments of the non-wetland emissions groups contributions from many source processes, mainly anthropogenic, which makes them difficult to interpret at the domain-scale. In the Nordics, the LUMIA inversions infer a 11% (LUMIA-Lprior) and 20% (LUMIA-Lpost) reduction of the non-wetland monthly emissions from January to June. In July both inversions point to a modest increase (+4-7%) compared to the prior, after which the emissions are progressively reduced, to reach near zero towards the end of the year. This is unrealistic, however the amplitude of the non-wetland emissions reduc-

tion is too large to just result from a flux misattribution: if re-attributed entirely to the wetland category, this would lead to (significantly) negative wetland emissions. A possible alternative (or complementary) explanation could be error in the CAMS boundary condition: the background concentrations ($y_{bg}$ in Equation 2) explains nearly 100% of the observed mixing ratio on several days towards the end of the year, especially at Hyytiälä (SMR) and Birkenes (BIR). The situation is also similar in the two sensitivity inversions.

### 3.2.4 Spatial distribution

The emission adjustments inferred in GRaB-AM and LUMIA optimizations are shown in Figure 8 (and SI Figure 2 for the sensitivity runs). To facilitate the comparison, wetland adjustments in LUMIA-Lpost are shown relative to the unoptimized LPJ-GUESS estimate (LPJ-GUESS-unopt).

The spatial distribution of wetland emission adjustments is very similar in the five data-informed products, and largely

proportional to the LPJ-GUESS-unopt emission estimate itself. The long error-correlations imposed on the LUMIA inversions (and intrinsic to the GRaB-AM optimization), combined with the relative concentration of wetland emissions in Northern Europe, ensure a convergence between the localization of flux corrections.

Among the most marked features in the adjustment to the "non-wetland" category, we note a doubling of the emissions in the Bretagne region of France, and in the Northern part of the Netherlands. This could point to underestimated agricultural

emissions, which are important in these two regions. Another marked feature is an important ($\approx 80\%$) reduction of the emissions in Northern Italy, which is well correlated with both high natural emissions (mainly geological) and high agricultural emissions.

We, however, need to ascertain a level of care when interpreting these emission adjustments: For instance emissions in the west of the continent can also result from the need to correct an inaccurate boundary condition. There can also be compensating

effects between adjustments of emission hotspots, such as the city of Paris or the Po Valley, and their surroundings. These emission corrections should be investigated, but fall outside the scope of our study.

### 3.3 Fit to observed data

A classical diagnostic in data assimilation is to compare results (optimized emissions or concentrations) to independent measurements. For GRaB-AM, such a validation has been conducted in Kallingal et al. (2024). For atmospheric inversions, the





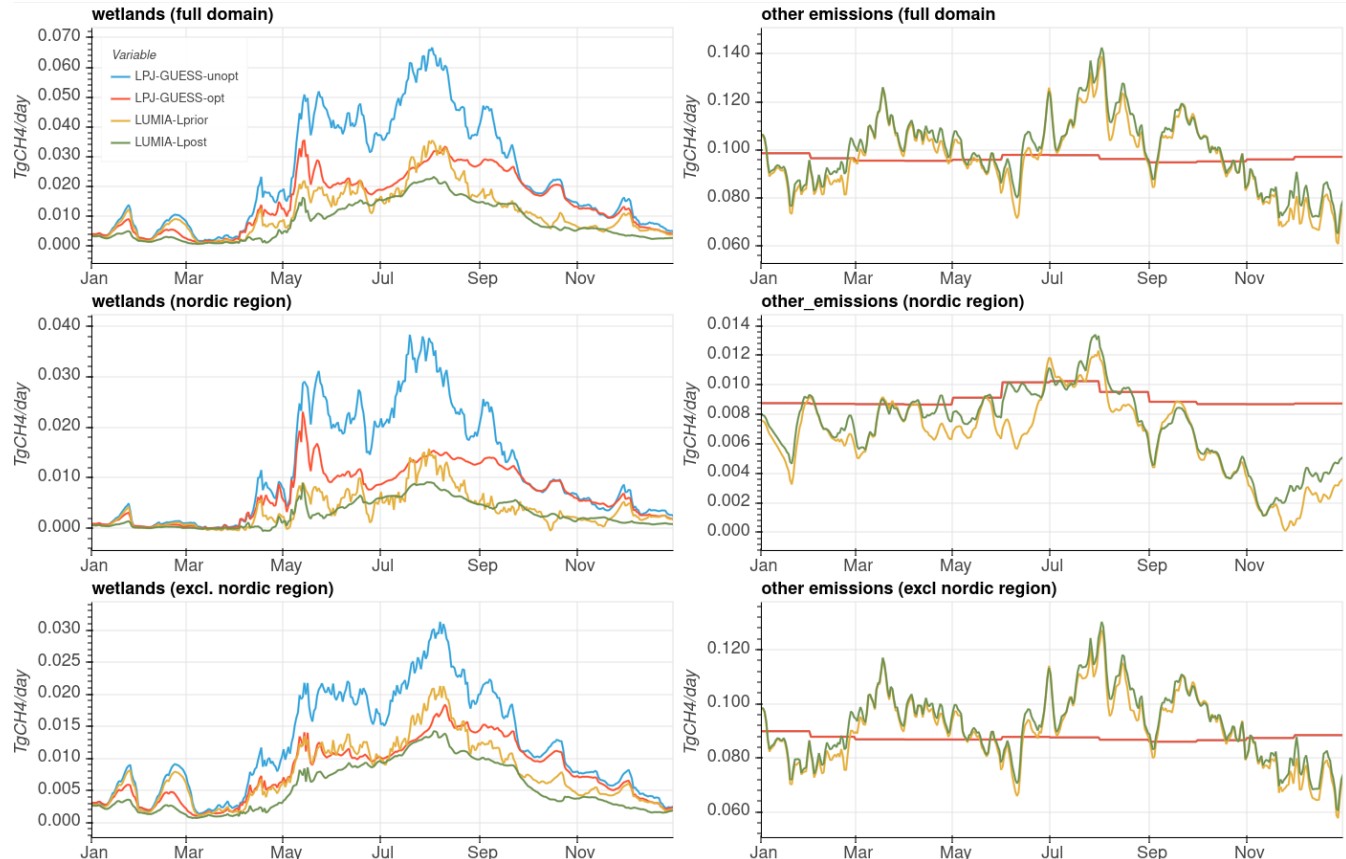

**Figure 7.** Daily emissions in the entire domain (top), in the Nordic region (middle) and outside it (bottom), for the wetland (left) and non-wetland (right) $CH_4$ emissions.

comparison is generally made with independent observations of the atmospheric composition, keeping in mind that biases due to the transport model would likely affect similarly the fit to assimilated data and to validation data.

For this study however, most of the available data in the Nordics has been assimilated, either in the LUMIA inversions (for concentration data), or in the GRaB-AM assimilation (for in-situ flux measurements). The aim of the model-data comparisons in this section is therefore not to derive an objective metric of the respective qualities of each emission estimates, but rather to gain insights on the forcings that lead the two optimizations system to adjust the emissions the way they did.

### 3.3.1 Eddy-covariance flux estimations

$CH_4$ emissions can be estimated locally on wetland scales using flux measurement techniques such as Eddy-covariance measurements, which involve capturing the covariance between the vertical wind speed and the concentration of methane, providing high-resolution data on gas exchange over wetlands. Such observations are for instance provided by the ICOS network in Eu-





**Figure 8.** Emission adjustments for wetlands (left) and non-wetlands (right) in the three data-informed simulations, compared to the unoptimized LPJ-GUESS model (LPJ-GUESS-unopt).





**Table 3.** percentage RMSE reduction, compared to the LPJ-GUESS-unopt site simulation, at the three eddy-covariance sites represented in Figure 9

| Simulation | Siikaneva | Degero | Zarnekow |
|---|---|---|---|
| LPJ-GUESS-opt (site) | 93 % | 98 % | 60 % |
| LPJ-GUESS-unopt (gridded) | -140 % | 43 % | 38 % |
| LPJ-GUESS-opt (gridded) | 43 % | 83 % | 45 % |
| LUMIA-Lprior | -3 % | 66 % | 21 % |
| LUMIA-Lpost | 93 % | 91 % | 40 % |
| LUMIA-Lprior+corr | -20 % | 67 % | 35 % |
| LUMIA-Lpost+corr | 77 % | 91 % | 30 % |

rope (ICOS RI et al., 2024), and the FLUXNET-CH4 dataset globally (Pastorello et al., 2020; Delwiche et al., 2021), which offers aggregates of high-quality $CH_4$ fluxes from wetlands. These networks try to offer a comprehensive coverage of the different types of wetlands (with differences in physiological features such as hydrology, soil characteristics, vegetation types, etc., and in spatial features, such as geographical distribution, size, landscape position and topography).

However, a direct comparison with gridded emissions is difficult, as the latter accounts for average conditions over the grid
cells, which can be very different from the local ones. This is illustrated in Figure 9, which shows a comparison between our four main emission estimates and in-situ flux measurements at three sites in the Nordic region (Zarnekow is slightly outside the Nordic domain used for the emission comparisons, see Figure 1), along with the site-level LPJ-GUESS simulations which were used to train the GRaB-AM optimization (see Section 2.1.2).

The fit of the modelled emissions against the observations is improved for most sites (clearly shown for e.g. Siikaneva and
Degero) but since the GRaB-AM optimization is seeking for an optimal parameter set fitting multiple sites simultaneously it is not surprising that there are still larger differences between simulated emissions and observations for any given individual site (as is the case for Zarnekov where the calibrated LPJ-GUESS model fails to simulate the observed peak values during August 2018).

The site-level simulations achieve systematically a better fit to the observations than the gridded products. Among the
gridded products, the best fit is obtained by LUMIA-Lpost, at Siikaneva and Degero, with RMSE reduction above 90% (Table 3), whereas the error reduction is lower at Zarnekow, with all the data-informed product in a 40% to 45% error reduction range. The two sensitivity inversions using ensemble-derived covariances lead to slightly worse fit than the base LUMIA inversions. We also note a tendency of the LUMIA inversions using LPJ-GUESS-unopt as a prior to infer significant negative emissions on some days (SI figure 4): LUMIA adjusts the emissions but preserves most of their original day-to-day variability, which
results in days with negative emissions.





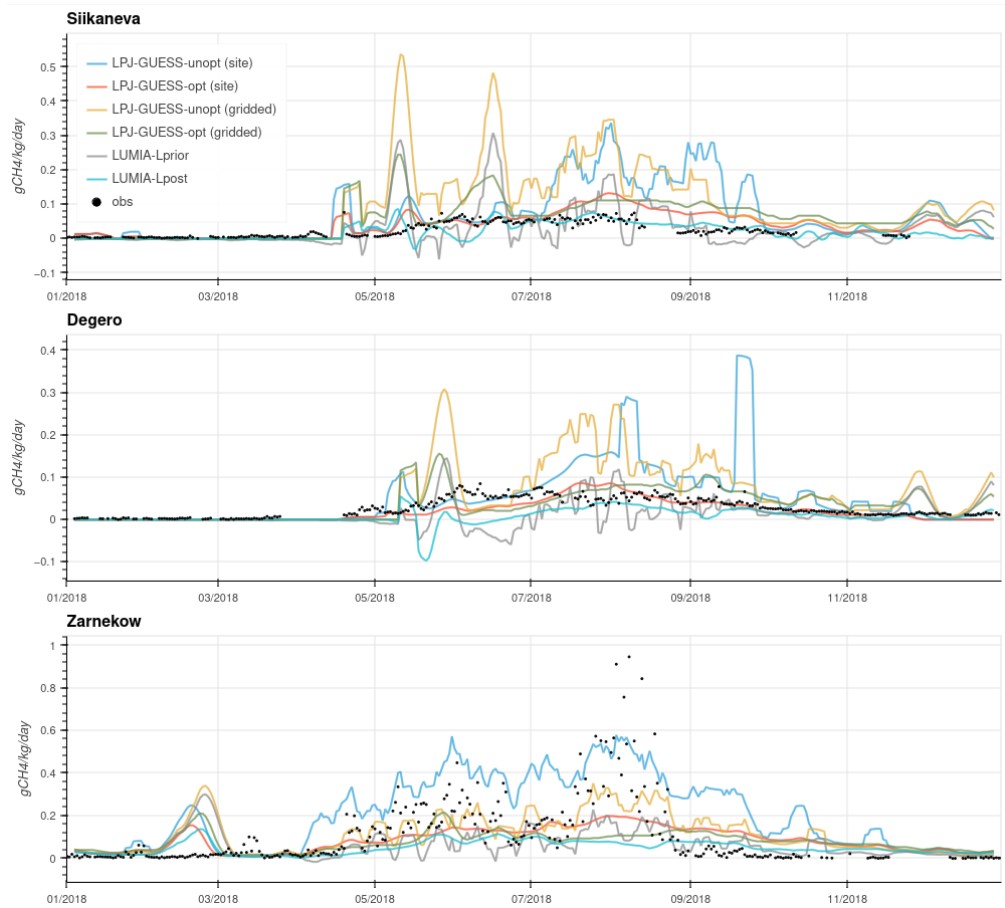

**Figure 9.** Modelled (solid lines) and observed (dots) methane emissions at three sites within the Nordic region. For clarity of the figure, a weekly rolling average has been applied to the modelled data. A version of this figure without smoothing can be found in supplementary materials.

### 3.3.2 Atmospheric CH$_4$ observations

The CH$_4$ concentrations corresponding to the four methane emission estimates are shown in Figure 10, for the six sites in the Nordic region. These sites are also among the ones where the relative contribution of wetland emissions to the foreground concentrations (i.e. the part of the concentrations that can be adjusted by LUMIA) is the highest. For figure clarity, the modelled timeseries have been averaged weekly. SI Figures 4 and 5 show the data without weekly averaging.

LPJ-GUESS-unopt leads to a strong overestimation of the concentrations throughout the summer (with a mean bias of up to 40.9 ppb at Norunda, and peak model-data mismatches exceeding 300 ppb). The fit obtained using LPJ-GUESS-opt emissions is much more in line with the observations, with mean biases ranging from -1.8 ppb at Hyltemossa to 6.6 ppb at SMR. However, the CH$_4$ concentrations are still significantly overestimated during the summer months, in particular at Norunda and Pallas.



**Table 4.** RMSE (in ppb) of the LUMIA simulations, for the six sites shown in Figure 10

| Simulation | Hyltemossa | Utö | Pallas | Birkenes | Norunda | Hyytiälla |
|---|---|---|---|---|---|---|
| LPJ-GUESS-unopt | 26.00 | 39.69 | 80.26 | 25.50 | 92.95 | 46.97 |
| LPJ-GUESS-opt | 25.22 | 34.44 | 39.30 | 21.42 | 54.10 | 38.21 |
| LUMIA-Lprior | 19.23 | 15.76 | 27.63 | 15.92 | 30.52 | 18.06 |
| LUMIA-Lpost | 19.13 | 15.32 | 15.90 | 15.57 | 15.64 | 16.17 |

The LUMIA inversions lead to comparable results in terms of mean bias, but LUMIA-Lpost performs better in terms of RMSE, in particular at Norunda and Pallas (Table 4). At these two sites, the emissions inferred in LUMIA-Lprior lead to modelled $CH_4$ concentrations well below the background values on some days. This is a consequence of the negative $CH_4$ emissions inferred by that inversion, already discussed in Section 3.3.1

The better fit of the LUMIA inversions is expected, since they assimilated these data. However, the overestimation of the observations in the LPJ-GUESS-unopt simulation is very large and a clear indication that the emissions modelled by the non-optimized LPJ-GUESS in the summer are refuted by the atmospheric observations. Other sources of uncertainties (transport model error, uncertainty on the background concentrations, uncertainty in non-wetland emissions) don't seem large enough to account for such an overestimation of the observed data.

# 4 Discussion

At the global level, methane emissions are relatively well constrained by the observed growth rate of $CH_4$ background sites, despite uncertainties on the magnitude and trends of the OH sink (Turner et al., 2019). However, significant uncertainties remain on regarding both anthropogenic and natural emissions in bottom-up approaches (Saunois et al., 2020). Concerning wetlands emissions specifically, the bottom-up estimates lead to a 40% to 50% uncertainty range (Kirschke et al., 2013; Melton et al., 2013). Inverse modeling can be used to infer constraints on the emissions at large scales (Bruhwiler et al., 2014; Houweling et al., 2017), but can only robustly constrain the net methane emissions, since this is what atmospheric $CH_4$ observations are sensitive to. Satellite observations, such as TROPOMI XCH4 retrievals (Nesser et al., 2024; Tsuruta et al., 2023), or retrievals from the upcoming CO2-M (Sierk et al., 2021) or MERLIN instruments (Ehret et al., 2017), can help increase the resolution at which the emissions are optimized, but their coverage is not constant (cloudiness, short day length at high latitudes in winter, etc.), and their signal-to-noise ratio is lower than that of in-situ observations for detecting emissions (because satellite $XCH_4$ retrievals quantify the column-averaged $CH_4$ mixing ratio, therefore they incorporate a stronger background contribution than surface observations). Some implementations of the inverse approach use observations of the isotopic composition of atmospheric methane ($\delta^{13}C$-$CH_4$, $\delta D$-$CH_4$) as a constraint on the source process distribution (Basu et al., 2022; Thanwerdas et al., 2024; Drinkwater et al., 2023), but the low amount of available data and the uncertainties on the isotopic signatures of methane emissions have limited the practicality of that approach.



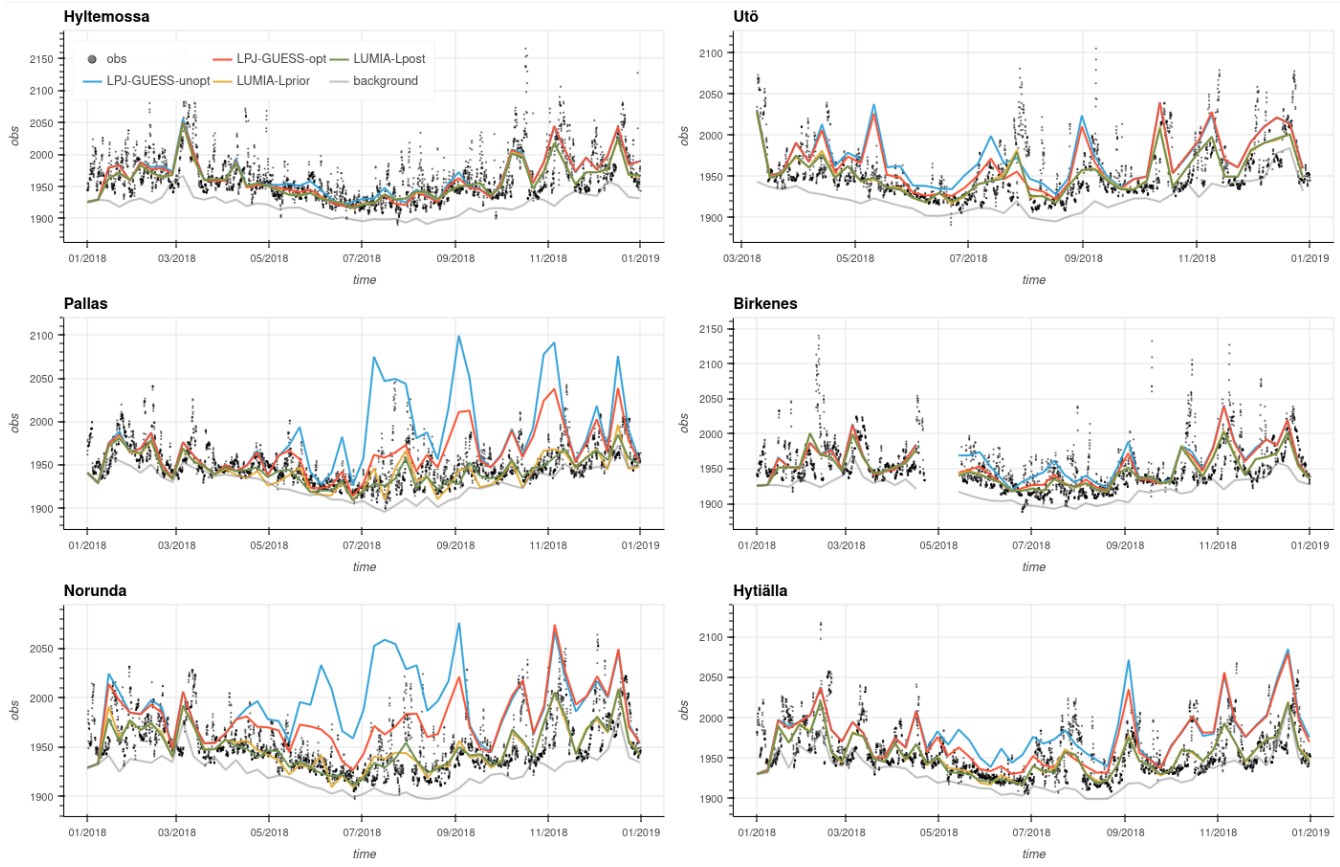

**Figure 10.** Modelled (solid lines) and observed (dots) CH$_4$ mixing ratio at the six observation sites within the Nordic region. Modelled time series are shown with as weekly rolling average. A non-smoothed version of the figure can be found in supplementary materials.

Our study combines two data-informed, in principle complementary: the DA approach (GRaB-AM) is process-specific and can lead to improvements in the prognostic capabilities of the underlying process model (LPJ-GUESS), but remains subject to possible large scale biases, both because of the lack of representativity of the assimilated data and the inaccuracy of LPJ-GUESS. The inverse approach (LUMIA) s arguably more reliable at large scales, but lacks spatial and process resolution.

   In this context, the development of a full CH$_4$ emission data assimilation system (CH4-DAS), combining a vegetation
and an atmospheric transport model and capable of assimilating both eddy-covariance measurements and atmospheric CH$_4$ observations appears as the next logical step. Such systems have been developed successfully for CO$_2$ (Rayner et al., 2005) and have shown promising results. For methane however, the development is complicated by the need to account for non-wetland methane emissions, which although less uncertain in relative terms, dominate the emission and emission uncertainty budget in absolute terms (Saunois et al., 2020), and by the complexity of wetland models, which can be highly non linear
(Kallingal et al., 2024).





As an intermediate solution, our study explores a two-step approach, with an atmospheric inversion informed by emission estimates and error correlations from a $CH_4$ DA approach. In the following section we further discuss the potential and limitations of both approaches, and how they can help us improve the LPJ-GUESS model.

### 4.1 LPJ-GUESS parameter estimation (GRaB-AM)

The GRaB-AM approach aims at optimizing specifically wetland emissions, by fitting the emissions of a DGVM (LPJ-GUESS) to eddy-covariance (EC) measurements. Indeed, the assimilation of EC yields a clear improvement in the fit not only to ecosystem data, but also to atmospheric $CH_4$ observations. The resulting emission estimates inherits the spatio-temporal structure from the process parametrizations implemented in the model. This ensures that the emissions remain consistent with their assumed relationships to factors such as climate and environmental forcings. Overall, it should improve the predictive capacity of

the model but can also lead to large systematic errors if the aforementioned parametrizations are insufficiently accurate (such as assumption of zero wind speed above wetlands, the lack of detailed representation of ebullition, and inadequate representation of wintertime emissions in the case of LPJ-GUESS), and/or if the sites used for training are not representative enough. A specific difficulty encountered in GRaB-AM is the high non-linearity of LPJ-GUESS which makes it very challenging to design a minimization algorithm that avoids getting local minima and/or parameter equifinality issues.

The comparison to atmospheric data however shows that LPJ-GUESS is likely still overestimating $CH_4$ emissions, even after having assimilated EC measurements. This could be related to the lack of emissions from wet mineral soils in LPJ-GUESS, which hence are attributed to wetlands emissions in this LPJ-GUESS version. The DA (GRaB-AM) improves on this but still results in too high emissions. This essentially points to required improvements in the model formulation such as for example for the somewhat simplistic representation of ebullition in LPJ-GUESS.

### 4.2 Atmospheric inversion (LUMIA)

This study is the first application of LUMIA inversions to a non-$CO_2$ tracer. Compared to the latest $CO_2$ applications Munassar et al. (2023); Gómez-Ortiz et al. (2023), the inversion setup has been simplified: the inversions adjust the emissions directly, instead of offsets to the prior emissions in these studies. This is permitted by the comparatively lower temporal resolution of the methane emissions. The uncertainties are of two orders: First, inversions rely on a transport model to establish the link

between observed $CH_4$ mixing ratio and emissions, which can bring systematic errors. Secondly, the source attribution of the emission adjustments depends for a large part of the prescribed emission uncertainties and error correlations.

The original aim was to construct the wetland emission error-covariance matrices based on the variability of the LPJ-GUESS ensembles. However, the ensemble correlations could not be (in a practical manner) approximated and inverted for using in LUMIA (in part because the spatial error correlations are not constant through time). We therefore opted for a more conven-

tional approach to constructing the error-covariance matrices, using the ensemble variability only to distribute a prescribed annual uncertainties in time and space. Results within the Nordic region of interest were very similar to those obtained in the "+corr" sensitivity runs (at least LUMIA-Lpost+corr), which is an indication that, in this region, the observations provide robust enough constraints on the emissions.



The uncertainty associated to transport is difficult to assess independently. Comparisons to independent (i.e. non assimilated) observations rely on the same model to estimate the link between concentrations and emissions, therefore they don't constitute a totally independent validation of the source-concentration relationships themselves. Model intercomparisons, such as TRANSCOM for global models (Gurney et al., 2004), and EUROCOM for regional models (Monteil et al., 2020) can help identifying divergences between models and inversion approaches. A detailed intercomparison was conducted between the LUMIA and CarboScope-Regional (CSR) inversion systems to quantify the importance of model biases in regional $CO_2$ inversions (Munassar et al., 2023), which highlighted a stronger sensitivity to emissions in LUMIA. While this could lead to overestimating the methane concentrations (given the correct emissions), the amplitude of the mismatches in LPJ-GUESS-unopt, and the fact that they occur specifically in regions with important wetland emissions rather plead for a significant overestimation of the methane emissions by LPJ-GUESS.

Both inversions pointed to a reduction of non wetland emissions in the Nordic region towards the end of the year, which seems unrealistic: fossil-fuel emissions (which dominate non-wetland emissions in the Nordic region) are not expected to display such intra-annual variability. However, it cannot be entirely a mis-attribution error, since the reduction in non-wetland emissions is an order of magnitude larger than the (posterior) wetland emission themselves. We noted that the CAMS background concentrations were very close to (or even higher than) the observed values in winter, especially at Pallas and Hyytiälä (Figure 10): as widespread negative emissions of $CH_4$ are unrealistic, this points to an overestimation of the background concentrations by the CAMS concentration baselines. This could lead to a widespread bias in the inferred emissions, but it is difficult to determine whether it affects the whole inversion period or just the winter months.

Some inversion systems allow the boundary condition to be adjusted (e.g. Steiner et al. (2024)), but this is risky in the absence of a proper quantification boundary condition uncertainty and of its variability. Here, the non-wetland emission category partly acts as a bias correction, but we must acknowledge this issue as a remaining source of uncertainty.

## 4.3 Refined estimate of European methane emissions at high latitude

We have refined the wetland emission estimate in the Nordic region to a range of 1.1 $TgCH_4$/year (LUMIA-Lpost) to 2.5 $TgCH_4$/year, significantly down from 4.3 $TgCH_4$/year in the original LPJ-GUESS-unopt estimate. While the difference between the LPJ-GUESS-unopt and the other estimates is large, the relative qualities of the three data-informed products are more difficult to assess.

The inversions lead to an improved representation of atmospheric observations, while not significantly degrading the fit to EC data compared to LPJ-GUESS-opt. Furthermore, we can speculate that a portion of the flux corrections to Nordic non-wetland emissions inferred by the inversions should in fact be attributed to wetlands: the comparison with in-situ flux measurements shows a tendency of LPJ-GUESS to model emissions in winter which are not confirmed by observations. Emission peaks are modelled in December at Siikaneva and Degero, and in March at Zarnekow (and are likely much more widespread since they are visible in the domain-aggregated emission timeseries shown in Figure 7), but are not present in the EC data. The inversions correct these peaks, but in part through an adjustment of the non-wetland emissions. On the other hand, the potential issues



with the background concentrations, highlighted in the previous section, could indicate an underestimation of the emissions in our inversions.

Wetland emission estimates from Arctic wetlands derived from six different vegetation models were compared in Aalto et al. (2024). They found that LPJ-GUESS was clearly at the high end of the range but not a complete outlier. Our inversion-optimized estimates, on the other hand, are well in line with the average of that ensemble. In Kallingal et al. (2024), the GRaB-AM optimization adjusted emissions from wetlands located above 45 degrees latitude from 43.09 TgCH$_4$/year to 37.54 TgCH$_4$/year. While the inversion part of the study concerns only a small portion of that domain, the reasonable agreement between LPJ-GUESS-opt and the LUMIA inversions reinforces our confidence in the GRaB-AM results.

## 4.4 Towards a coupled flux-concentration CH$_4$ data assimilation system

The initial aim of the study was the implementation of a two-step estimation approach, with a transmission of uncertainty between the flux DA (GRaB-AM) and the atmospheric inversion (LUMIA) parts. Two main complications were encountered: first, the error structure from the DA step could not be easily approximated in a form usable by the inversion (i.e. as a set of standard deviations and spatial and temporal correlation matrices). This purely technical limitations could be overcome, e.g. by using an ensemble minimization approach in LUMIA, which usually don't need an explicit representation of the error covariance matrices (e.g. Bisht et al. (2023)).

A more fundamental issue is the complexity of the LPJ-GUESS model (and of dynamic global vegetation models in general), which makes their optimization against eddy-covariance data arduous (e.g. Famiglietti et al. (2021)). This could be overcome by the development of diagnostic models for wetland methane emissions, similar to those existing for CO$_2$ (Mahadevan et al., 2008; Knorr and Heimann, 1995; Potter et al., 1993).

In the past years, subsequent work has also been conducted by the anthropogenic emission inventory compilers to produce uncertainty estimates (e.g. Solazzo et al. (2021)), which should lead to a better representation of the anthropogenic emission uncertainties in inversions, and in turn improve the reliability of their source attribution.

## 5 Conclusions

We have implemented European CH$_4$ inversions using the LUMIA inversion system, making use of wetland emission estimates and associated uncertainties from a CH$_4$ emission data assimilation (DA) system (GRaB-AM), based on the LPJ-GUESS model Kallingal et al. (2024). We focused our analysis on the estimation of wetland emissions in the Nordic region, since wetlands emissions are the most uncertain term in the methane budget, and since they dominate the emission budget in that region.

We compared several data-informed wetland emission estimates: DA of eddy-covariance flux measurements (GRaB-AM), inversion constrained by atmospheric CH$_4$ observations (LUMIA), and inversion constrained by atmospheric observations, and, indirectly, by eddy-covariance data (through the EC data informed GRaB-AM prior). All simulations clearly point to a strong (by a factor 2 to 3) overestimation of the CH$_4$ emissions by the LPJ-GUESS model. The GRaB-AM approach leads to significant improvement of the fit to atmospheric data (which it didn't assimilate), which constitutes a form of additional



validation for the approach. The inversion using emissions from the GRaB-AM data assimilation as a prior also leads to the best overall fit to observations.

We have explored using model-based error covariances in the LUMIA inversions, to improve their capacity to resolve contributions of wetlands to the total methane emissions. The impact was relatively negligible within the Nordic region, but more significant in regions where wetlands contribute a smaller fraction of the uncertainty. Making full use of these model-based error covariance would require adaptations in the inversion approach (by e.g. using an ensemble-based optimization technique). It also implies relying on the specific wetlands $CH_4$ emissions processes as implemented in the LPJ-GUESS DGVM. Nonetheless, these obstacles could be overcome, and our study shows the potential for a joint flux-concentration $CH_4$ data assimilation system, which we will explore in future studies.

*Code and data availability.* The source code for this project, as well as a selection of the data (fit to observations, monthly prior and posterior emissions) will be made available on a public repository once the study is accepted. The full datasets and code can be obtained by contacting the main author.

*Competing interests.* The contact author has declared that none of the authors has any competing interests

*Author contributions.* GM designed the LUMIA inversion system and performed the atmospheric inversions. JTK designed the GRaB-AM data assimilation system and computed the LPJ-GUESS ensembles. GM, MS and JTK collectively designed the study. GM wrote the manuscript, with contributions and critical feedbacks from MS and JTK.

*Acknowledgements.* We thank Sander Houweling and Liesbeth Florentie for coordinating the inverse modeling intercomparison that lead to this study, preparing and distributing the inversion inputs (prior emissions, observations, background concentrations). We also thank Arjo Segers (TNO) who computed the CAMS simulation and the baseline concentrations used in the inversions. We acknowledge all the data providers cited in Table 2. We acknowledge the PIs of the in situ flux measurements obtained from FLUXNET (https://fluxnet.org/data/fluxnet2015-dataset/) and AVAA-SMEAR (https://smear.avaa.csc.fi/, for SMEAR II data, Ivan Mammarella and colleagues) for the open data.

This research has been partly supported by the Strategic Research Area "Biodiversity and Ecosystem services in a Changing Climate" (BECC), Lund University (grant no. DnrV 2018/467), and is a contribution to the Strategic Research Area "ModElling the Regional and Global Earth system" (MERGE). BECC and MERGE are funded by the Swedish government. This research has also been partly supported by the "CoCO2" (grant no. 958927) and the "AVENGERS" (grant no. 101081322s) projects from the European Union's Horizon 2020 and Horizon Europe Framework Programmes for Research and Innovation, funded by the European Union.

The computations and data handling were enabled by resources provided by the National Academic Infrastructure for Supercomputing in Sweden (NAISS), partially funded by the Swedish Research Council through grant agreement no. 2022-06725.



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
