# Peer review of "CH4 emissions from Northern Europe wetlands: compared data assimilation approaches"

_EGUsphere, 2024_

## Referee Comment (RC2)

**Review on Monteil et al., egusphere-2024-3122**

**General comments**

This paper studies Nordic wetland $CH_4$ emissions from the LPJ-GUESS process model and the LUMIA atmospheric inverse model. In the example year (2019), they found good improvements with atmospheric $CH_4$ observations over Europe, modelled by FLEXPART Lagrangian particle dispersion model, after optimizations using GRaB-AM flux data assimilation framework, LUMIA, and the combination of both data assimilation systems. Although the authors did not fully couple the flux and atmospheric data assimilation systems, the results look promising. I found that the way they examined the process model estimates also using atmospheric transport models is valuable, which has not been a very common approach. Ideally, additional data assimilation (i.e. in addition to EC data, also assimilating atmospheric data) would improve regional emission estimates, providing improved fit to data to both EC and atmospheric data. This was not yet archived in this study, and possibly because the coupling was not done fully. I anyway look forward to see their development to fulfil the goal. In addition, improving uncertainty estimates from the process models and using such information in the atmospheric inverse models are advanced approach, which should be explored more. This will also improve source attribution of emission estimates from atmospheric inverse models, which is a hot-topic now. Therefore, this paper presents an important work that contributes significantly to the carbon cycle studies.

In general, I have a few points which could be discussed more.

- The assimilated data are different, not only the quantity but also regarding number of sites/observations. How much of the differences between the optimization approaches are due to this? What if you only assimilate atmospheric data from the stations that are close to the EC sites, do you think you can still archive same improvements?

- I understand that it is difficult to examine which processes contribute to the over- or underestimation of emissions from the atmospheric inverse model. However, now you have attempted to combine the information from the process model and its parameter estimation, could you further discuss what is the problem that causes the process model based estimates to be over- or underestimated even after assimilating the EC data?

- As you mentioned briefly, validation of this model seems to be challenging. After assimilating both EC and atmospheric data, there are not much validation data left. But what data are available and would be the optimal or good way to evaluate the results? How can you say that end results are (or will be) better than only assimilating one or the other?

- Based on your results, would you argue that we need both EC and atmospheric data to estimate regional wetland $CH_4$ emissions, or only one or the other is enough? Can you identify an optimal way of designing observation network – should we have EC and atmospheric measurements at the same locations, or mostly in different locations?

Please also revise following specific and technical points before being accepted for publication.

**Specific comments**

Abstract: I am not sure if you need all the references. Could you try to write without references? If some spaces are left after revision, could you possibly add a bit more about results?

L3: "poor spatial and process resolution" and L413: "process resolution"
I am not sure what you mean by "process resolution". Could you rephrase a bit?

L5: "leading to more realistic emission estimates."
Would it be better to say e.g. "realistic emission distribution"? You said earlier that inversion approach can provide robust total emissions.

L11-13: "In this study, we used the LUMIA regional atmospheric inversion system (Monteil and Scholze, 2021) to confront wetland emissions from the GRaB-AM approach to atmospheric CH4 measurements in Europe. We then perform inversions using the information from GRaB-AM as prior."
Does the second sentence repeating what is said in the first sentence? Could you consider to merge these sentences?

L23: "for the global methane budget"
Do you perhaps mean global natural methane budget? On overall, I think anthropogenic emissions contribute more than tropical wetlands.

Section 2.1.2
Please provide information about EC sites in LUMIA domain. Although a full list can be found in Kallingal et al. (2024), it is reader friendly to repeat the information also in this paper.

L108-109: Why do you use different ranges for prior and posterior ensembles? How are these percentages decided?

L160-161: CAMS provides two inversion estimates, one using surface air-samples, and another using also satellite data. Which product did you use?

L171-172: "All the emissions were regridded from their original resolution to the 0.25°, daily resolution of the FLEXPART footprints."
What method did you use in case original resolution is coarser than 0.25° or daily?

L175: "influence on observations"
I guess here you mean atmospheric observations. This is true, but would it be better if you say e.g. "contribution to seasonality of total emissions" so that it is connected to emissions rather than atmospheric states.

L178: "the fact that the observation network is relatively dense in Northern Europe"
It is probably not only about how the dense the observation network is, but also locations and surroundings of the sites. If all the sites are close to anthropogenic sources, the constraint on wetland sources well be weak. Please clarify this point a bit more.

L187: "they are set proportional to the absolute value of the emissions"
Can you specify how much?

L198-199: Did you use all the data available, or did some filtering based on observation quality, quantified by flags? From which dataset did you collect these from?

L201-202: ICOS data provides various measurement uncertainty. What variable did you use? 20 ppb measurement uncertainty sounds rather high. What is the rational for the choice?

Section 3.1: Please consider modifying "for each emission category" to "for wetland emission category" in L227 if you wish to describe only about wetland emission uncertainties in Section 3.1. Please anyway add somewhere in text about description of how you calculated uncertainties for non-wetland emission category.

L214-215: From the description here, I assumed $\varepsilon^i_{mod}$ changes in time as $n_{obs}$ vary weekly, but in Table 2, only a single "model error" value is given for each site. Could you explain how you end up with one value per site?

L321-323: How much uncertainties did you put to the background concentrations? What makes SMR and BIR special that $y_{bg}$ explains 100% of observed mixing ratios? If it only matters on several days in the end of year, can it explain the large reduction that starts already in the beginning of November?

L350: "the way they did."
I do not quite understand this last part. Maybe the sentence can be slightly revised?

L360-363: Did you take into account wetland area (i.e. how large are wetlands) within the grid cell? If so, did you take the wetland area information from LPJ-GUESS?

L370: I see the largest reduction in RMSE in LPJ-GUESS-opt (site) rather than LUMIA-Lpost.

L371: Zarnekow has observed emission peak in August, where models failed to capture. This is probably the main reasons why RMSE did not improve. What is this associated with, and why models failed to capture the peak?

L345: "overestimating CH4 emissions" → "overestimating regional CH4 emissions"?

L434-439: Is it only due to lack of emissions from wet mineral soils? How about overestimation of wetland emissions in regions where EC data are not available, e.g. due to wrong representation of wetland area, or biases in climate forcing data?

L443-444: "lower temporal resolution of the methane emissions"
I do not fully agree. Do you mean temporal variations of methane emissions are weaker in the sense that it does not have strong diurnal cycle? Please rephrase.

L476-479: How are these estimates compared to other studies? Is it a common feature that emissions are reduced from process-based model estimates after optimizations (either by assimilating EC or atmospheric data)? If so, does it mean that some significant improvement is needed in the process models related to descriptions of processes? You may combine with last paragraph of 4.3 and expand discussion a bit more? I would also be interested to know about seasonality – how is it compared to other models/estimates?

L480-481: I guess ideally, you'd have even better fit to EC data with atmospheric inversion. Why do we fail to do so?

L501-505: Could you kindly elaborate more on what you mean by the complexity of the vegetation model (i.e. complexity in what sense) and "diagnostic models" (i.e. what are you going to diagnose)?

L506-508: As you showed in this study, anthropogenic emissions play an important role in estimation of regional $CH_4$ fluxes in Europe as their magnitude is larger than wetland emissions. So this development will be crucial. What are the uncertainties in anthropogenic $CH_4$ emissions, and how large/small are they compared to the wetland emission uncertainties? If you implement the information, what would been the consequent results? Is there any atmospheric inversion studies (possibly of $CH_4$) that employ different uncertainty estimates for different sources?

Figure 1.
- Please add latitude/longitude coordinate labels of the grids.
- Please check coordinates of the EC sites. I wonder what is a site on the Baltic sea…?
- I do not see Siikaneva with green star. Is it just overlapped with red star? Please make it clear (also for Zarnekow).
- Please add/rephrasing the caption a bit to make it clear what type of observations you are talking about:
  - LUMIA inversion domain (cyan grid), the position of the **atmospheric observation sites** used in the LUMIA inversions... The red dots mark the position of the **eddy-covariance** sites used in Section 3.3.1.
- Labels: I would prefer to use "sites" than "obs", and modify "$CH_4$ obs" to e.g. "Atmospheric sites" to make it clear (EC also measures $CH_4$).

Figure 2:
- Are these regional emissions for the LUMIA or Nordic domain?
- Unit in the y-axis should be Tg $CH_4$ day$^{-1}$, with 4 being subscript.
- What is bmb?
- Could you kindly add wetland emissions used in LUMIA-Lpost inversion?

Figure 3:
- Is this annual averages? Please specify in caption.
- The spatial resolutions in the panels seem to be different. Please consider using the same resolution in all.
- Why the maximum colour scale for wetland sources does not have arrow as in others? Is it on purpose?

Figure 4:
- Unit in the figure on left is missing.
- I guess the colour label on the right figure should not be emissions, but emission uncertainties.

Figure 5: bottom panel: please use same labels as in the top panel

Figure 6: Is it possible add uncertainty ranges?

Figure 8: Do I understand it correctly that wetlands LUMIA-Lpost panel (left middle) show differences compared to LPJ-GUESS-unopt? How do the differences between LUMIA-Lpost and LPJ-GUESS-opt look like?

Figure 9: Could you use different colour for LIMIA-Lpost? It's a bit hard to distinguish it from LPJ-GUESS-unopt.

Figure 10:
- Are these observed values hourly data, and are they all assimilated in LUMIA?
- How about plotting observations also as weekly averages? It is now hard to compare between observations and modelled values as they have different temporal resolution.

Table 1:
- Please check resolution columns. Temporal/spatial resolution columns seem to be mixed up.

- Column on "climatological" can be filled with "no" also. I wonder if that matters here, though, as the study is only for one year.
- Why spatial(?) resolution of geology and termites are missing?

Table 2:
- Is the values "model error" $\varepsilon^i_{mod}$, i.e. not the quadratic sum of measurement and model uncertainties, i.e. diagonals of R, but just model uncertainties? If so, could you also add e.g. range of measurement uncertainties as well?
- nobs seems to exceed number of observations per year (365 days x 7 hours (11:00-17:00 LT) = 2555). Could you explain (somewhere in text) why? Is the information about e.g. spin-up spin-down missing in the description of inversion setup?
- Please add units for latitude, longitude, elevation and inlet height.
- Lan et al. reference seems to be missing in Bibliography. Please also remember to add publication year for this reference.

Table 3:
- Do I understand it correctly that RMSE here means RMSE in differences between simulated and the EC measurements? Please make this clear. Could you also add the RMSE in LPJ-GUESS-unopt site simulation?
- Is it correct that negative values mean increase in RMSE?

Table 4: Could you also add biases?

**Technical comments**
Unit "/year" should be written as year$^{-1}$

In general, overuse of parentheses could be improved. Short ones are ok, but I would prefer to write them as proper sentences in many cases. Please consider revising.

Please check reference styles. For example, "(e.g. Rayner et al. (2005))" should be written as "(e.g. Rayner et al., 2005)".

don't / doesn't / didn't → do not / does not / did not

L109: "time" → "times"?

L147: Please provide references to FLEXPART.

L166: The term super-category is only used here. Can you just say categories?

L173: Please mention Figure 2 first.

L174: "that has exhibits": there are two verbs?

L186: $\sigma_w$ → $\sigma_w$ is not defined yet. do you mean $\sigma_{xc}$?

L191: "in-situ observations from 43 European in-situ and flask measurement sites, "
"in-situ" is repeated. Should it be rephrased as e.g. "continuous and discrete observations from 43 European in-situ sites"?

L192: "ICOS network of in-situ measurements"

I would prefer to say "ICOS network of atmosphere stations."

L194: "and the free ,"
Some words missing?

L232-233: "were estimated by through two" → "estimated by" or "estimated through"

L275: Figure 5 → do you mean Figure 6?

L292-293:
"roughly a halving" → "roughly a half of"
"mainly during the" → "mainly due to reduction of emissions during"?

L350: "two optimizations system" → "two optimization systems"

L356: "CH4 fluxes from wetlands" → "CH4 flux measurements *(or data or observations)* from wetlands"

L364: "improved for most sites" → "improved for most sites after optimizations"

L395-409: I would prefer to have this in Introduction as this is a kind of general description.
L410: "two data-informed, in principle complementary:" → "two data-informed, in principle complementary, approaches:

L413: Remove "s" after "(LUMIA)"

L434: "and/or" → "or"

L476-477: Please add simulation name that estimated 2.5 Tg $CH_4$ $year^{-1}$.

L511-501: "based on the LPJ-GUESS model Kallingal et al. (2024)" → "based on the LPJ-GUESS model by Kallingal et al. (2024)" or "based on the LPJ-GUESS model (Kallingal et al., 2024)"

L517: overestimation of the **wetland** $CH_4$ emissions?

L525: "emissions processes" → "emission processes"

Acknowledgement: Please add acknowledgement for the ICOS data. https://www.icos-cp.eu/how-to-cite

---

## Author Comment (AC1)

We thank both reviewers for their positive and constructive feedback on the manuscript. We agree with most of their comments and have used them to significantly improve the paper.

Reviewer #2 commented on the abnormally high number of observations assimilated in the LUMIA inversions, which led us to discover that one observation filter was not correctly applied. To account for this, we had to recompute all the LUMIA inversions, which has led to revisions of the figures and of the results section. However, the new results do not fundamentally change the outcome of the study (if anything, the range of estimates has narrowed down).

Please find below our detailed answers (with the reviewers comment in blue, and our answers in black).

**Review 1**

*Line 1: Abstract gives methodological overview but lacks statement about the outcomes (what did you found) of this study.*

We have revised the entire abstract, in response to comments from both reviewers

*Line 117: LUMIA (Monteil and Scholze, 2021)*

Fixed

*Line 126: 33°N (not S?)*

Fixed

*Line 145: What do you mean for "qualitatively equivalent resutls"*

Differences exist between the results, but these are rather negligible and don't affect the interpretation of the results. We have replaced "qualitatively equivalent" by "similar".

*Line 227-231: This part should be moved to Method*

In fact, this is already in the methods (Section 2.2.4), but we think that briefly recalling it here helps understanding the following paragraphs. But the text should have been pointing to Section 2.2.4 and not 2.2.1. This has been corrected.

*Table2: At several stations (hpb, hun, mhd, pal), two records are shown. What is the difference (e.g., in-situ or flask)?*

Yes, these correspond to sites that have both flask and in-situ data. There are very few flask data in comparison to in-situ data, so their impact on the inversions is negligible (and the observation uncertainty is anyway inflated according to the number of observations per week at a given site, so the observational constraint isn't really increased). They were kept in the workflow as they can have an importance in places where there is only flask data, although such a situation doesn't occur in the experiments presented in this manuscript.

*Line 249–254: This part is not about result but about methodology.*

This is an analysis of the ensembles of LPJ-GUESS simulations computed following the method described in Section 2.1.2.

*Line 290: May (not may).*

corrected

*Line 352–357: This part may be moved to Discussion.*

This sentence is indeed not a result. However, this section shows model-data comparisons, which could easily be mis-interpreted: some of the observations have been assimilated in some experiments but not in others, there is differences in the scale of representativeness of the observations (local flux measurements compared to grid-cell averages), etc. The first two paragraphs aim at clarifying that context.

**Review 2**

*Abstract: I am not sure if you need all the references. Could you try to write without references? If some spaces are left after revision, could you possibly add a bit more about results?*

We have removed one reference and summarized the findings in the abstract.

*L3: "poor spatial and process resolution" and L413: "process resolution" I am not sure what you mean by "process resolution". Could you rephrase a bit?*

We have revised the entire abstract

*L5: "leading to more realistic emission estimates." Would it be better to say e.g. "realistic emission distribution"? You said earlier that inversion approach can provide robust total emissions.*

We have revised the entire abstract

*L11-13: "In this study, we used the LUMIA regional atmospheric inversion system (Monteil and Scholze, 2021) to confront wetland emissions from the GRaB-AM approach to atmospheric CH4 measurements in Europe. We then perform inversions using the information from GRaB-AM as prior." Does the second sentence repeating what is said in the first sentence? Could you consider to merge these sentences?*

The first sentence referred to the propagation of GRaB-AM emissions with LUMIA (i.e. CH4-LPrior), while the second one referred to their further optimization (CH4-LPost). We however agree that it wasn't very clear so we haven't used these sentences when re-writing the abstract.

*L23 "for the global methane budget" Do you perhaps mean global natural methane budget? On overall, I think anthropogenic emissions contribute more than tropical wetlands.*

yes, we meant the budget of wetland emissions. We have fixed the sentence as suggested.

*Section 2.1.2 Please provide information about EC sites in LUMIA domain. Although a full list can be found in Kallingal et al. (2024), it is reader friendly to repeat the information also in this paper.*

In Kallingal et al., 2024, GRaB-AM accounts for observations at fourteen sites, in the Northern Hemisphere high latitudes. Sites outside the LUMIA domain (e.g. in North America) don't contribute less than those inside to the parameter estimation. Therefore, if we are to provide information on these sites, we need to do it for all of them, which would end up repeating large chunks of Kallingal et al., 2024. Therefore, we prefer to stick with just referring to that paper.

*L108-109: Why do you use different ranges for prior and posterior ensembles? How are these percentages decided?*

The ranges are actually not different, but the formulation was confusing. In both cases, the samples are drawn from their 90% confidence interval. But in the "prior" (LPJ-GUESS-unopt) ensemble, the PDFs are non Gaussian, to avoid non-sensical negative values for some parameters, therefore the relation between confidence interval and standard deviation is more complicated.

We have adjusted the text, slightly.

*L160-161: CAMS provides two inversion estimates, one using surface air-samples, and another using also satellite data. Which product did you use?*

It relies on the inversion assimilating surface observations. This now has been clarified.

*L171-172: "All the emissions were regridded from their original resolution to the 0.25°, daily resolution of the FLEXPART footprints." What method did you use in case original resolution is coarser than 0.25° or daily?*

Nothing particular, the low resolution emissions were just projected on a higher resolution grid, so the emissions in kgCH4/m2 at a given lat/lon/time remain the same

*L175: "influence on observations" I guess here you mean atmospheric observations. This is true, but would it be better if you say e.g. "contribution to seasonality of total emissions" so that it is connected to emissions rather than atmospheric states.*

Indeed, it's better this way. We have adjusted the text according to the reviewer's suggestion

*L178: "the fact that the observation network is relatively dense in Northern Europe" It is probably not only about how the dense the observation network is, but also locations and surroundings of the sites. If all the sites are close to anthropogenic sources, the constraint on wetland sources well be weak. Please clarify this point a bit more.*

This is precisely what the previous sentence refers to ("There is also a geographical separation between emissions from wetlands …"). The immediate surrounding of the sites is not that important (CH4 is a long-lived tracer, so the observations are sensitive to the emissions aggregated over a very large area, and the ICOS sites are chosen not to be in the direct vicinity of emission hotspots).

*L187: "they are set proportional to the absolute value of the emissions" Can you specify how much?*

No, at least not in a meaningful way. They are initially set to 100% of the absolute values of the emissions, but as mentioned in the following sentence (l188 of the original manuscript), the aim is just to set the contribution of each variable to the overall uncertainty. They are later multiplied by a uniform scaling factor ($\gamma$) in Eq. 3, to achieve a category-specific target annual uncertainty (reported in Table 1). We have clarified this in the revised manuscript.

*L198-199: Did you use all the data available, or did some filtering based on observation quality, quantified by flags? From which dataset did you collect these from?*

We used all the data available in the dataset. The observation dataset was provided through the CoCO2 project (Ioannidis et al., 2025), we did not apply any further filtering (except the time of the day selection).

*L201-202: ICOS data provides various measurement uncertainty. What variable did you use? 20 ppb measurement uncertainty sounds rather high. What is the rational for the choice?*

As mentioned in the previous answer, the data comes from a curated dataset prepared for previous inversion intercomparison exercises (e.g. Ioannidis et al., 2025, Thompson et al., 2021}, and contains one single measurement error column. However, the manuscript was not correct: 20 ppb is the default value (in case none is provided), not the minimum one. The rationale for such a high value was to avoid over-constraining the inversions with data of unknown quality. But after verification, it turns out that although the setting was switched on, it was not actually used, since all valid observations had a valid model error value in the dataset. We have clarified the origin of the data and removed the reference to the 20 ppb value from the text.

*Section 3.1: Please consider modifying "for each emission category" to "for wetland emission category" in L227 if you wish to describe only about wetland emission uncertainties in Section 3.1. Please anyway add somewhere in text about description of how you calculated uncertainties for non-wetland emission category.*

We have slightly modified the start of the section, to clarify that this is just a reminder of the method, described in more detail in Section 2.2.4 (which describes the uncertainty in a more generic way, for other categories too). We have also modified the title of section 3.1, to clarify that it refers only to wetlands

*L214-215: From the description here, I assumed εimod changes in time as nobs vary weekly, but in Table 2, only a single "model error" value is given for each site. Could you explain how you end up with one value per site?*

The value reported in Table 2 is $\sigma_{mod}^{site}$ . We have clarified this in the caption of Table 2.

*L321-323: How much uncertainties did you put to the background concentrations? What makes SMR and BIR special that ybg explains 100% of observed mixing ratios? If it only matters on several days in the end of year, can it explain the large reduction that starts already in the beginning of November?*

The background concentrations are prescribed in our inversions, based on a CAMS inversion constrained with surface observations. However, the CAMS inversion doesn't assimilate all the observation sites used in our study. It is therefore possible that it incorrectly represents some of them. It would in principle be possible for us to further optimize the background concentration to account for this, but without a clear idea of how large the uncertainty on this should be, the outcome would be very difficult to interpret. We

therefore chose to simply prescribe it and acknowledge possible shortcomings of the approach.

*L350: "the way they did." I do not quite understand this last part. Maybe the sentence can be slightly revised?*

We have slightly reformulated the sentence.

*L360-363: Did you take into account wetland area (i.e. how large are wetlands) within the grid cell? If so, did you take the wetland area information from LPJ-GUESS?*

Yes, the emissions are expressed in gCH4 per day and square meter of wetland area, with the wetland area information from LPJ-GUESS. We have added that information to the caption of Figure 9, and also corrected the unit labels in that figure, which were incorrect (gCH4/m2/day instead of gCH4/kg/day).

*L370: I see the largest reduction in RMSE in LPJ-GUESS-opt (site) rather than LUMIA-Lpost.*

Indeed, but "LPJ-GUESS opt (site)" is not a gridded product but run at site-level (the full sentence is "Among the gridded product, the best fit …").

*L345: "overestimating CH4 emissions" → "overestimating regional CH4 emissions"?*

corrected

*L434-439: Is it only due to lack of emissions from wet mineral soils? How about overestimation of wetland emissions in regions where EC data are not available, e.g. due to wrong representation of wetland area, or biases in climate forcing data?*

We have removed the reference to the lack of emissions from wet mineral soils, which, although not wrong, is clearly not the explanation for the overestimated emissions here. We have reformulated that paragraph and introduced the hypothesis of errors in the wetland area.

*L443-444: "lower temporal resolution of the methane emissions" I do not fully agree. Do you mean temporal variations of methane emissions are weaker in the sense that it does not have strong diurnal cycle? Please rephrase.*

Indeed, we have replaced "resolution" by "variability"

*L476-479: How are these estimates compared to other studies?*

Direct comparisons with other studies are difficult because of the limited extent of our study domain. We have however expanded that section with a comparison to the JSBACH-HIMMELI mode, for which we had data since it was used in Ioannidis et al., 2025.

*Is it a common feature that emissions are reduced from process-based model estimates after optimizations (either by assimilating EC or atmospheric data)?*

We don't really understand the question. Inversions adjust the prior emissions, so, yes, it is common that emission estimates are reduced by the inversions (and it is common as well that they are increased). Maybe the reviewer is asking whether there is a general tendency to reduce them? Not to our knowledge.

*If so, does it mean that some significant improvement is needed in the process models related to descriptions of processes?*

At least in LPJ-GUESS, yes. And we acknowledge this in the manuscript.

*You may combine with last paragraph of 4.3 and expand discussion a bit more? I would also be interested to know about seasonality – how is it compared to other models/estimates?*

We have expanded the discussions, in particular with a comparison to the JSBACH-HIMMELI model. Direct comparisons with other studies are cumbersome, because of the differences in study area and temporal extent, which requires re-downloading and processing the data, therefore we had to limit it to what was already available to us.

*L480-481: I guess ideally, you'd have even better fit to EC data with atmospheric inversion. Why do we fail to do so?*

Inversions are only indirectly (through their prior) constrained by EC data, and the major biases have already been corrected by GRaB-AM in LPJ-GUESS-opt, therefore there is not much scope for improvements in inversions using that product as a prior. Furthermore, the EC data represents emissions at one specific observation site, while the LUMIA emissions are representative of one 0.25°x0.25° grid cell, so even in a perfect world, we wouldn't expect a full match between the two.

The text of Section 4.3 has been adapted to better reflect this.

*L501-505: Could you kindly elaborate more on what you mean by the complexity of the vegetation model (i.e. complexity in what sense) and "diagnostic models" (i.e. what are you going to diagnose)?*

By diagnostic models, we refer to models that attempt to estimate the CH4 emissions based on empirical relationships to a set of observable parameters, rather than full DGVMs that attempt to simulate the full carbon cycle.

We have further developed this paragraph in the revised manuscript.

*L506-508: As you showed in this study, anthropogenic emissions play an important role in estimation of regional CH4 fluxes in Europe as their magnitude is larger than wetland emissions.*

Note that our focus in this study is on wetland emissions specifically. Anthropogenic emissions are important as well, but better addressed by other studies, e.g. Ioannidis et al., 2025, to which we participated, with a slightly different setup (JSBACH-HIMMELI wetland emissions instead of LPJ-GUESS).

*So this development will be crucial. What are the uncertainties in anthropogenic CH4 emissions, and how large/small are they compared to the wetland emission uncertainties?*

The ratio of uncertainties between anthropogenic and wetland emissions depends a lot on the location. At the scale of Europe, we attributed a total uncertainty of 5 TgCH4 to the "non-wetland" emission category, which is largely dominated by anthropogenic emissions, vs. 0.5 TgCH4/year for wetlands. But over the Nordic region, on which we focused our analysis, this ratio is a lot more balanced (the prior wetland and non-wetland emissions are on the same order of magnitude, for the Nordic region, but, assuming that wetland emissions are a lot more uncertain, they dominate the uncertainty budget.

*If you implement the information, what would been the consequent results?*

We are not really sure what the reviewer refers to by "implement the information". As mentioned, our inversions already account for uncertainty in anthropogenic emissions (in fact, for the whole "non-wetland" category, but it is largely dominated by anthropogenic emissions), but using a rather arbitrary estimation for that uncertainty, which limits our trust in the source-attribution of the emission adjustments by the inversions, especially in regions where both anthropogenic and natural emissions are significant (which is why we didn't really focus on other European wetlands, such as those in the Netherlands, Great Britain or Ireland).

Having a more objective quantification of the anthropogenic emission uncertainties would strengthen our confidence in our results, but it would probably not dramatically change them (unless our current estimate of these uncertainties is too incorrect).

*Is there any atmospheric inversion studies (possibly of CH4) that employ different uncertainty estimates for different sources?*

Our inversions do technically employ different uncertainty estimates for different sources ("wetlands" and "non-wetlands") and there are many other examples, for CH4 and for other tracers, including some rather old ones (e.g. Meirink et al., 2007). The difficulty is not

to set up inversions resolving emissions per category, but to obtain robust results with them.

Implementing more realistic anthropogenic emission uncertainties (including correlations) and resolving the emissions for finer categories is of course necessary once they become available, but it will really only be useful in conjunction with the assimilation of more observations (e.g. satellite data, isotopic data, flux observations, etc.), which is the more technically challenging part.

*Figure 1:*

*Please add latitude/longitude coordinate labels of the grids.*

We have added them in the revised figure

*Please check coordinates of the EC sites. I wonder what is a site on the Baltic sea…?*

It was a typo in the coordinates of Degero (all EC sites are plotted in green, and the ones used in Fig 9 are overplotted in red, but somehow, the green point for Degero was shifted South).

*I do not see Siikaneva with green star. Is it just overlapped with red star? Please make it clear (also for Zarnekow).*

Yes, the idea was to have a red star on top of the green one, to mark the sites covering the year 2018. But clearly it was not readable enough. We have adjusted the size of the green and red stars to make it more visible, and have clarified the caption.

*Please add/rephrasing the caption a bit to make it clear what type of observations you are talking about: "LUMIA inversion domain (cyan grid), the position of the atmospheric observation sites used in the LUMIA inversions… The red dots mark the position of the eddy-covariance sites used in Section 3.3.1."*

See reply above.

*Labels: I would prefer to use "sites" than "obs", and modify "CH4 obs" to e.g. "Atmospheric sites" to make it clear (EC also measures CH4).*

We have followed the reviewer's suggestion and edited the labels.

*Figure 2:*

*Are these regional emissions for the LUMIA or Nordic domain?*

They are for the LUMIA domain. We have included a cleaner revised figure.

*Unit in the y-axis should be Tg CH4 day-1, with 4 being subscript.*

See comment above

*What is bmb?*

bmb is for biomass burning. We have clarified this in the revised figure.

*Could you kindly add wetland emissions used in LUMIA-Lpost inversion?*

The emissions of the LUMIA-Lpost inversion are shown in Figure 7. We think it woud be confusing to have them in the methods section.

*Figure 3:*

*Is this annual averages? Please specify in caption.*

Yes, these are annual averages. We have now included that information in the caption

*The spatial resolutions in the panels seem to be different. Please consider using the same resolution in all.*

The spatial resolution is the same in all panels (0.25°), but the original datasets were at different resolutions (very low for the geological emissions), see revised Table 1.

*Why the maximum colour scale for wetland sources does not have arrow as in others? Is it on purpose?*

It's more or less on purpose (it's a feature of the plotting library). For wetlands, all grid cells are within the range of the colourbar, while for the other categories, there are grid cells that exceed it, therefore the colourbar is shown with an arrow. Since these are very localized, extending the color range to accommodate them would make the rest of the plots look very flat, therefore we chose that compromise.

*Figure 4:*

*Unit in the figure on left is missing.*

We have fixed this in the revised manuscript. We have also changed the plotted units from nmolCH4/m2/s to gCH4/m2/day, which is more common, therefore the range of the colourbar has changed.

*I guess the colour label on the right figure should not be emissions, but emission uncertainties.*

The colour label is "uncertainty reduction (%)" and not "emissions", therefore we don't understand this comment.

*Figure 5: bottom panel: please use same labels as in the top panel*

We have fixed this in the revised manuscript.

*Figure 6: Is it possible add uncertainty ranges?*

No, not easily, at least not in a meaningful way. The inversion system does compute an estimate of the uncertainty reduction, but its interpretation can be quite misleading (it's a diagnostic of the inversion system, not a proper uncertainty estimate: it doesn't account for uncertainties on prescribed components of the inverse problem (e.g. the transport model, the boundary conditions, the type of correlations used, etc.), therefore it's not a value that we like to communicate, especially when mixing several data assimilation systems. We think that a much better measure of uncertainty comes from intercomparison exercises, such as Ioannidis et al., 2025, for instance.

*Figure 8: Do I understand it correctly that wetlands LUMIA-Lpost panel (left middle) show differences compared to LPJ-GUESS-unopt? How do the differences between LUMIA-Lpost andLPJ-GUESS-opt look like?*

All panels show differences vs. LPJ-GUESS-unopt (there are too many possible comparisons to show them all, we opted to compare everything to the un-optimized LPJ-GUESS. We have however now included the comparisons to LPJ-GUESS-opt in Figure S2).

*Figure 9: Could you use different colour for LIMIA-Lpost? It's a bit hard to distinguish it from LPJ-GUESS-unopt.*

We tried to make the figures as readable as possible, but with six timeseries on the same plot, it is hardly avoidable that some colors will look alike. We now made the figure full-width, which hopefully should make it more legible.

*Figure 10:*

*Are these observed values hourly data, and are they all assimilated in LUMIA?*

Yes, the observed values are hourly, but only afternoon observations are assimilated in LUMIA (see Section 2.2.5).

*How about plotting observations also as weekly averages? It is now hard to compare between observations and modelled values as they have different temporal resolution.*

The main take away point from this figure is the overestimated concentrations in the LPJ-GUESS-unopt and LPJ-GUESS-opt cases (i.e. without further optimization by LUMIA). This is sufficiently visible on the figure. The differences between the two LUMIA simulations and

the observations are, by construction, rather negligible (since LUMIA adjusts the emissions precisely to fit these observations).

In the revised manuscript, we have reduced the smoothing windows to 48 hours, which is a better compromise between showing the information and retaining legibility. We have also removed the smoothing from the background concentration timeseries (since it's not needed for legibility), which facilitates discussion on the role of the background.

*Table 1:*

*Please check resolution columns. Temporal/spatial resolution columns seem to be mixed up.*

Indeed, we have fixed this

*Column on "climatological" can be filled with "no" also. I wonder if that matters here, though, as the study is only for one year.*

It doesn't matter in the sense that we are not studying inter-annual variability, but it still matters if there is a particular emission anomaly that year, that would not be captured by the climatological emission field.

*Why spatial(?) resolution of geology and termites are missing?*

The original data comes from a collection of point sources, therefore without a resolution as such. However, the fields we received had been pre-regridded, and clearly to a rather low resolution (4th panel in Figure 3), therefore we agree that Table 1 is somewhat misleading. We have revised the table and its caption.

*Table 2:*

*Is the values "model error" $\varepsilon_{imod}$, i.e. not the quadratic sum of measurement and model uncertainties, i.e. diagonals of R, but just model uncertainties? If so, could you also add e.g. range of measurement uncertainties as well?*

Model uncertainty here refers to the weekly model representation error assumed at each site. As noted in response to an earlier comment, the measurement error is typically much smaller, and rather constant from site to site. The main objective of that column is highlighting how we adjust the relative weight of each observation site.

*nobs seems to exceed number of observations per year (365 days x 7 hours (11:00-17:00 LT) = 2555). Could you explain (somewhere in text) why? Is the information about e.g. spin-up spin-down missing in the description of inversion setup?*

Thanks a lot for being so thorough! There was indeed a preprocessing error, some of the night-time data had not been correctly filtered (see main reply). The numbers in the new table are correct.

*Please add units for latitude, longitude, elevation and inlet height.*

We have added them.

*Lan et al. reference seems to be missing in Bibliography. Please also remember to add publication year for this reference.*

We used the references provided along with the datasets, ideally with a DOI and a year when available, but not all of them provide it. Lan et al. is present in the bibliography (line 720 of the original manuscript).

*Do I understand it correctly that RMSE here means RMSE in differences between simulated and the EC measurements? Please make this clear. Could you also add the RMSE in LPJ-GUESS-unopt site simulation?*

The table shows "RMSE reduction, compared to the LPJ-GUESS-unopt site simulation", not "RMSE compared to the EC measurements": 100 * (RMSE(LPJ-GUESS-unopt) - RMSE(sim)) / RMSE(LPJ-GUESS-unopt). We have clarified this in the caption.

*Is it correct that negative values mean increase in RMSE?*

yes

*Table 4: Could you also add biases?*

yes, we have added them.

**Technical comments**

*Unit "/year" should be written as year-1*

We leave this for the publisher to fix, once the paper is accepted.

*In general, overuse of parentheses could be improved. Short ones are ok, but I would prefer to write them as proper sentences in many cases. Please consider revising.*

This comment is difficult for us to address, since it is not specific.

*Please check reference styles. For example, "(e.g. Rayner et al. (2005))" should be written as "(e.g. Rayner et al., 2005)".*

See our response to the first technical comment.

*don't / doesn't / didn't → do not / does not / did not*

See our response to the first technical comment.

*L109: "time" → "times"?*

fixed

*L147: Please provide references to FLEXPART.*

Indeed, we have added the correct reference.

*L166: The term super-category is only used here. Can you just say categories?*

We think it would be confusing here. The word "super-category" and the sentence that follows are self-explanatory enough.

*L173: Please mention Figure 2 first.*

Figure 2 is mentioned at the next line (174) …

*L174: "that has exhibits": there are two verbs?*

It should have been "that exhibits". We have fixed this.

*L186: σw → σw is not defined yet. do you mean σxc?*

yes, we have fixed this.

*L191: "in-situ observations from 43 European in-situ and flask measurement sites, "in-situ" is repeated. Should it be rephrased as e.g. "continuous and discrete observations from 43 European in-situ sites"?*

We have removed the first "in-situ"

*L192: "ICOS network of in-situ measurements" I would prefer to say "ICOS network of atmosphere stations."*

We have replaced by "ICOS network of atmospheric in-situ measurements"

*L194: "and the free ," Some words missing?*

yes, the word "troposphere" was missing

*L232-233: "were estimated by through two" → "estimated by" or "estimated through"*

we have removed "by"

*L275: Figure 5 → do you mean Figure 6?*

yes.

*L292-293: "roughly a halving" → "roughly a half of"*

we think "halving" is correct.

*"mainly during the" → "mainly due to reduction of emissions during"?*

we think our sentence is correct.

*L350: "two optimizations system" → "two optimization systems"*

The sentence has been reformulated, in response to another comment

*L356: "CH4 fluxes from wetlands" → "CH4 flux measurements (or data or observations) from wetlands"*

fixed

*L364: "improved for most sites" → "improved for most sites after optimizations"*

that's implied by the word "improved" (which refers to a before/after state)...

*L395-409: I would prefer to have this in Introduction as this is a kind of general description.*

We think it's relevant in the discussion.

*L410: "two data-informed, in principle complementary:" → "two data-informed, in principle complementary, approaches:*

fixed.

*L413: Remove "s" after "(LUMIA)"*

fixed (s -> is)

*L434: "and/or" → "or"*

it can be both, so and/or

*L476-477: Please add simulation name that estimated 2.5 Tg CH4 year-1.*

fixed

*L511-501: "based on the LPJ-GUESS model Kallingal et al. (2024)" → "based on the LPJ-GUESS model by Kallingal et al. (2024)" or "based on the LPJ-GUESS model (Kallingal et al., 2024)"*

fixed

*L517: overestimation of the wetland CH4 emissions?*

fixed

*L525: "emissions processes" → "emission processes"*

fixed

**References**

Ioannidis, E., Meesters, A., Steiner, M., Brunner, D., Reum, F., Pison, I., Berchet, A., Thompson, R., Sollum, E., Koch, F.-T., Gerbig, C., Wang, F., Maksyutov, S., Tsuruta, A., Tenkanen, M., Aalto, T., Monteil, G., Lin, H., Ren, G., Scholze, M., and Houweling, S.: An inter-comparison of inverse models for estimating European CH4 emissions, Earth System Science Data Discussions, pp. 1–42, https://doi.org/10.5194/essd-2025-235, publisher: Copernicus GmbH, 2025

Kallingal, J. T., Scholze, M., Miller, P. A., Lindström, J., Rinne, J., Aurela, M., Vestin, P., and Weslien, P.: Assimilating Multi-site Eddy-Covariance Data to Calibrate the CH4 Wetland Emission Module in a Terrestrial Ecosystem Model, EGUsphere, pp. 1–32, https://doi.org/10.5194/egusphere-2024-373, 2024b.

Meirink, J. F., Bergamaschi, P., Frankenberg, C., d'Amelio, M. T. S., Dlugokencky, E. J., Gatti, L. V., Houweling, S., Miller, J. B., Röckmann, T., Villani, M. G., and Krol, M. C.: Four-dimensional variational data assimilation for inverse modeling of atmospheric methane emissions: Analysis of SCIAMACHY observations, Journal of Geophysical Research: Atmospheres, 113, https://doi.org/10.1029/2007JD009740, 2008.

Thompson, R. L., Groot Zwaaftink, C. D., Brunner, D., Tsuruta, A., Aalto, T., Raivonen, M., Crippa, M., Solazzo, E., Guizzardi, D., Regnier, P., and Maisonnier, M.: Effects of extreme meteorological conditions in 2018 on European methane emissions estimated using atmospheric inversions, Philosophical Transactions of the Royal Society A: Mathematical, Physical and Engineering Sciences, 380, 20200443, https://doi.org/10.1098/rsta.2020.0443, 2021.